# Thermal Conductivity for p–(Bi, Sb)$_2$Te$_3$ Films of Topological Insulators

**Lidia N. Lukyanova \*, Yuri A. Boikov, Oleg A. Usov, Viacheslav A. Danilov, Igor V. Makarenko and Vasilii N. Petrov**

Ioffe Institute, Russian Academy of Sciences, 26 Politekhnicheskaya, St. Petersburg 194021, Russia
* Correspondence: lidia.lukyanova@mail.ioffe.ru

**Abstract:** In this study, we investigated the temperature dependencies of the total, crystal lattice, and electronic thermal conductivities in films of topological insulators p–Bi$_{0.5}$Sb$_{1.5}$Te$_3$ and p–Bi$_2$Te$_3$ formed by discrete and thermal evaporation methods. The largest decrease in the lattice thermal conductivity because of the scattering of long-wavelength phonons on the grain interfaces was observed in the films of the solid-solution p–Bi$_{0.5}$Sb$_{1.5}$Te$_3$ deposited by discrete evaporation on the amorphous substrates of polyimide without thermal treatment. It was shown that in the p–Bi$_{0.5}$Sb$_{1.5}$Te$_3$ films with low thermal conductivity, the energy dependence of the relaxation time is enhanced, which is specific to the topological insulators. The electronic thermal conductivity was determined by taking into account the effective scattering parameter in the relaxation time approximation versus energy in the Lorentz number calculations. A correlation was established between the thermal conductivity and the peculiarities of the morphology of the interlayer surface (0001) in the studied films. Additionally, the total $\kappa$ and the lattice $\kappa_L$ thermal conductivities decrease, while the number of grains and the roughness of the surface (0001) increase in unannealed films compared to annealed ones. It was demonstrated that increasing the thermoelectric figure of merit ZT in the p–Bi$_{0.5}$Sb$_{1.5}$Te$_3$ films formed by discrete evaporation on a polyimide substrate is determined by an increase in the effective scattering parameter in topological insulators due to enhancement in the energy dependence of the relaxation time.

**Keywords:** bismuth telluride; solid solutions; films; thermal conductivity; scattering parameter; topological insulator

## 1. Introduction

Currently, high-performance thermoelectrics based on bismuth and antimony chalcogenides with a tetradymite structure [1,2] are attracting much attention as promising 3D topological insulators (TIs). Topological surface states (TSS) in TIs arise due to the inversion of the energy gap edges caused by strong spin–orbit interaction [3–5]. In addition, the bulk becomes an insulator, and the electrons on the surface acquire unusual spin momentum locked with linear dispersion and spin polarization specific to the Dirac fermions [6,7]. The linear dispersion and tight interaction between spin and momentum, owing to time-reversal symmetry, prevent fermions from backscattering on non-magnetic defects. This promotes an enhancement in the mobility of charge carriers [8]. Such topological phenomena expand the application possibilities of TIs in various fields of physics [9–12], including thermoelectricity [4,5,8,13].

An enhancement in thermoelectric performance in TI films based on Bi$_2$Te$_3$ is associated with an increase in the energy dependence of the spectral distribution of the mean free paths of both phonons and electrons [8,13]. Nontrivial energy band structure and large spin–orbit coupling of Dirac fermions in topological insulators account for the enhancement in the energy dependence of the spectral distribution of the mean free path of the electrons $l_F(E)$ [4,5]. The estimations of $l_F(E)$ in TIs have shown that the electrons possess a wider spectrum than phonons. The effect of electron energy filtering in TIs [14,15] provides an

increase in the Seebeck coefficient due to the scattering of charge carriers on the grain interfaces in the films [4,13].

The strong dependence on energy of the mean free path of phonons in the energy range, which is significantly narrower than the electron energy range, accounts for the intensive phonon scattering on an interface between two grains in a polycrystalline material. Furthermore, the enhancement in the phonon scattering results in a decrease in the crystal lattice thermal conductivity in chalcogenide films [8,13,16,17]. In ref. [18], the effect of phonon scattering on grain interfaces on the reduction in the lattice thermal conductivity were considered in solid solutions based on bismuth telluride. Depending on features of the phonon spectrum, the acoustic phonons with low frequency and long wavelength have larger mean free path values than high-frequency ones [18].

The largest heat transfer in the films is determined by long-wavelength phonons that mainly affect the decrease in the lattice thermal conductivity $\kappa_L$ due to enhanced scattering. However, at low temperatures up to the Debye temperature ($T_D$ = 145 K), the decrease in $\kappa_L$ is determined mainly by phonons scattering on acceptor antisite defects and on impurity atoms in solid solutions [19,20]. Furthermore, an additional decrease in $\kappa_L$ occurs in the layered films of $Bi_2Te_3$ and its solid solutions when phonons are scattered on van der Waals interfaces in the superlattices with a period of about 2 nm, consisting of two inverted quintiples located between the Te(1) layers [21]. With increasing temperature, the effect of scattering on the $\kappa_L$ by point defects is reduced while the scattering of phonons on the grain interfaces becomes dominant. However, near room temperature, the contribution of phonon–phonon scattering becomes apparent [16,22–24].

A specific feature of topological thermoelectrics is the residual bulk electrical conductivity related to the presence of bulk defects [25,26]. The reduction in bulk conductivity occurs as a result of the optimization of the thermoelectric composition owing to mutual compensation of contributions of acceptor and donor intrinsic defects, providing the increase in TSS contribution to the total conductivity. However, in $Bi_2Te_3$-based films, the residual bulk conductivity cannot be completely eliminated.

The systematization of resistivity values in bismuth and antimony chalcogenide films allows one to distinguish, depending on the residual resistivity, two main directions of using TIs [3,5,27–29]. Materials with low values of residual resistivity are of high interest for property studies and applications in thermoelectricity [5,28,29]; similarly, materials with high values are significant in the research [27] and development of topological devices [3], as well as for enabling studies of relativistic phenomena, discussed in [30]. A quantitative assessment of these two types of TIs is presented in [27–29].

The listed properties of TIs determine a decrease in thermal conductivity and an increase in the thermoelectric power coefficient (Seebeck coefficient). The properties also provide an enhancement in the thermoelectric performance of chalcogenide films compared to bulk thermoelectrics despite a slight reduction in electrical conductivity. In this work, the effect of scattering mechanisms on total $\kappa$, lattice $\kappa_L$, and electronic $\kappa_e$ thermal conductivities was investigated. This effect is associated with the peculiarities of the energy dependence of the relaxation time in the submicron p–$Bi_{0.5}Sb_{1.5}Te_3$ and p–$Bi_2Te_3$ films obtained by discrete and thermal evaporation methods. The selection of formation technique and film composition allows for the optimization of the scattering mechanisms of phonons and electrons, which promote a decrease in the lattice thermal conductivity $\kappa_L$ due to the effect of the interlayer surface morphology, grain interfaces, and van der Waals superlattice.

## 2. Film Deposition Technique and Structure

The polycrystalline films of solid solutions p–$(Bi,Sb)_2Te_3$ and p–$Bi_2Te_3$ were obtained by the methods of discrete and thermal evaporation in an isothermal chamber which provides a vacuum of $1 \times 10^{-6}$ torr and homogeneous temperature distribution along the substrate plane. Fresh muscovite mica cleavage planes with a thickness of 3–10 μm and polyimide films with a thickness of 6–20 μm were used as substrates.

To obtain the films by the thermal evaporation method, the starting material was loaded into a quartz crucible heated by a molybdenum coil. The films deposition rate was 10–15 Å/s. During the film formation by discrete evaporation, the initial material in the form of powder with a grain size of about 10 μm was passed by small portions into heated quartz crucible, where it instantly evaporated.

The variation in temperature of the crucible and substrate material of mica (muscovite) and polyimide showed that the optimal temperatures to obtain films with the specified composition are 800–850 °C for the evaporator and 250–300 °C for the substrate. Table 1 represents the technological parameters of the formed p–$Bi_{0.5}Sb_{1.5}Te_3$ and p–$Bi_2Te_3$ films for thermal conductivity studies.

**Table 1.** The p–$Bi_{0.5}Sb_{1.5}Te_3$ and p–$Bi_2Te_3$ films formation technique parameters.

| No. | Formation Technique | Substrate | Heat Treatment | Thermoelectric Power Coefficient $\alpha$, μV K$^{-1}$ |
|---|---|---|---|---|
| | | | p–$Bi_{0.5}Sb_{1.5}Te_3$ | |
| 1 | discrete evaporation | polyimide | unannealed | 242 |
| 2 | discrete evaporation | polyimide | annealed | 215 |
| 3 | thermal evaporation | polyimide | unannealed | 200 |
| 4 | discrete evaporation | muscovite | annealed | 223 |
| | | | p–$Bi_2Te_3$ | |
| 5 | discrete evaporation | polyimide | unannealed | 234 |
| 6 | thermal evaporation | muscovite | unannealed | 203 |

To describe the structure of $Bi_2Te_3$ and its solid solutions, a primitive rhombohedral or hexagonal unit cell is used. According to X-ray powder diffraction data, the hexagonal unit cell of the $Bi_2Te_3$ has *a* and *c* parameters of 4.3805 Å and 30.487 Å, respectively [31]. The Te atoms possess either six Bi neighboring atoms or three Bi and Te atoms, allowing us to distinguish the sequence of simple layers and more complex formations consisting of five layers, called quintuples. The considered hexagonal unit cell contains three such quintuples. The Te and Bi atomic layers in the quintuple alternate in the sequence of (-Te(1)-Bi-Te(2)-Bi-Te(1)-), while the chemical bonds in the layers are mainly covalent with some degree of ionicity. It is known that in the p–$Bi_{0.5}Sb_{1.5}Te_3$ solid solutions, the Sb atoms substitute Bi. In these materials, intrinsic antisite point defects on the sites of tellurium $Bi_{Te}$ and impurity defects caused by Sb→Bi atom substitutions in solid solutions are revealed. The quintuple boundaries are the interlayer van der Waals surfaces or cleavage planes (0001). The quintuples are bonded by weak van der Waals forces, which provide a slight exfoliation of the crystal along the (0001) planes perpendicular to the crystallographic *c* axis. The layered structure of the $Bi_2Te_3$ crystals and solid solutions determines the significant anisotropy of the transport properties.

In the considered materials, the interlayer surface (0001) possesses the minimum value of free energy [32,33]. It facilitates the formation of stable crystal seeds of bismuth and antimony chalcogenides, which are formed with an orientation predominantly along the c axis perpendicular to the substrate plane, even on substrates with a large mismatch of the crystal lattice parameters. In the process of heat treatment at T = 390 °C in the Ar atmosphere, an intense selective evaporation of the Te atoms from the grain interfaces occurs, in addition to the secondary recrystallization. The depletion of the grain interfaces by tellurium affects the thermoelectric properties and in turn leads to an increase in the charge carrier concentration in the p–$Bi_{0.5}Sb_{1.5}Te_3$ and p–$Bi_2Te_3$ films [16]. It should be noted that studies on the morphology of the interlayer surface (0001) of p–$Bi_2Te_3$ films using conductive atomic force microscopy [34] showed an increase in local electrical conductivity in the region of the grain interfaces.

### 3. Atomic Force Microscopy Study of Interlayer Surface (0001) in the Films

The investigation of the interlayer surface (0001) morphology in the p–$Bi_{0.5}Sb_{1.5}Te_3$ and p–$Bi_2Te_3$ films was carried out using atomic force microscopy (AFM) in semicontact mode.

Figures 1–3 present typical two-dimensional (2D) and three-dimensional (3D) morphology images, and the profiles and the histograms of nanofragment distribution on the surface (0001) depending on the height for all studied samples. As an example, the p–$Bi_{0.5}Sb_{1.5}Te_3$ films were analyzed (samples 1 and 4, Table 1). In these films, the relief of the surface (0001) is composed of separate nanofragments, islands, terraces consisting of coalescent islands, and rows containing dislocations (Figure 1). The observed relief is formed by the diffusion processes and elastic stresses, which result in the deformation of the interlayer surface during the film formation. Separate nanofragments arising on the (0001) surface are scattering centers for phonons, and they lead to a decrease in the thermal conductivity of films based on chalcogenides of bismuth and antimony.

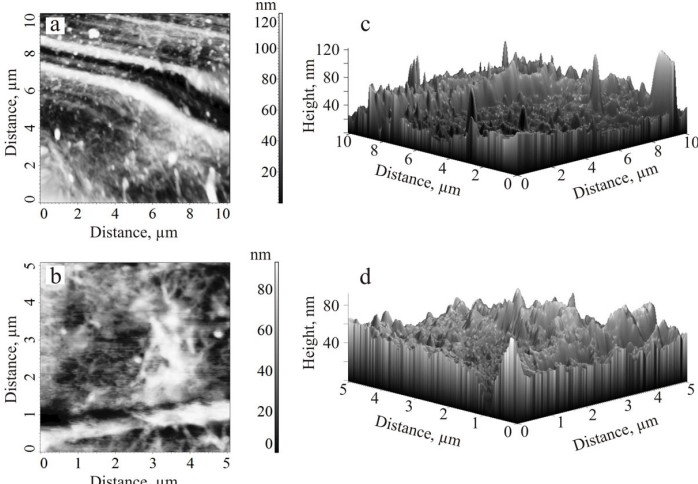

**Figure 1.** The (**a**,**b**) 2D and (**c**,**d**) 3D images of surface morphology (0001) of the unannealed p–$Bi_{0.5}Sb_{1.5}Te_3$ film obtained by discrete evaporation on a polyimide substrate (**a**,**c**) and of the annealed one deposited on a mica substrate (**b**,**d**).

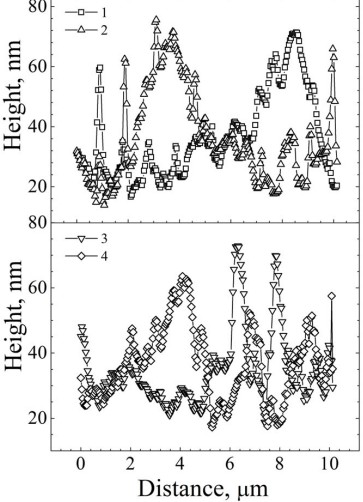

**Figure 2.** The profiles (1–4) of the surface morphology (0001) images of the p–$Bi_{0.5}Sb_{1.5}Te_3$ films (sample 1, Table 1) obtained along arbitrary and mutually perpendicular directions in Figure 1a: 1, 2 and 3, 4. Average height of nanofragments on the surface (0001) in nm: 1–36.35; 2–36.24; 3–35.54; 4–36.49.

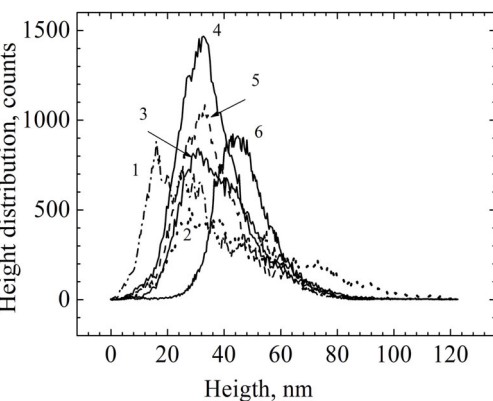

**Figure 3.** Height distribution of the nanofragments on the interlayer surface (0001) in the p–$Bi_{0.5}Sb_{1.5}Te_3$ films depending on the heights (1–3) in the annealed film obtained by discrete evaporation on a mica substrate and (4–6) the unannealed film deposited on a polyimide substrate.

The average heights of nanofragments on the surface (0001) (Figure 1a) are about 36 nm and have similar values for profiles obtained on various surface regions (Figure 2), indicating the homogeneity of the sample surface relief. Further information about the surface relief (0001) was found from the analysis of histograms (Figure 3), which were determined from the morphology images of the p–$Bi_{0.5}Sb_{1.5}Te_3$ films (samples 1 and 4, Table 1, Figure 1).

One can estimate the influence of annealing from the analysis of the distribution of nanofragments on the interlayer surface (0001) depending on their height, considering the p–$Bi_{0.5}Sb_{1.5}Te_3$ films obtained by discrete evaporation as an example (Figure 3). The number of nanofragments with sizes ranging from 16 to 28 nm (Figure 3, curves 1–3) was maximal for p–$Bi_{0.5}Sb_{1.5}Te_3$ film subjected to annealing (Table 1, sample 4). For the unannealed film, the dimensions of most nanofragments were from 31 to 45 nm (Table 1, sample 1, Figure 3, curves 4–6).

The average heights of nanofragments $R_a$ and the root mean square of height deviations of nanofragments $R_q$ (i.e., roughness) vary from 5.5 to 7 nm in the annealed p–$Bi_{0.5}Sb_{1.5}Te_3$ film, while in the unannealed film, $R_a$ and $R_q$ increase from 10 to 15.85 nm. Thus, the influence of annealing results in both a decrease in the number of nanofragments of maximum sizes and a decrease in $R_a$ and $R_q$ values compared with the unannealed p–$Bi_{0.5}Sb_{1.5}Te_3$ film.

Fourier images of the morphology of the film surface (0001) for annealed p–$Bi_{0.5}Sb_{1.5}Te_3$ deposited by the discrete evaporation on a mica substrate were obtained using the Fast Fourier Transform (FFT). These images represent the intensities distribution of two-dimensional reciprocal space being centered at the Γ point of the Brillouin zone (Figure 4a,b). The spectral components of intensities near the Brillouin zone center are associated with the interference of quasiparticle excitations of surface electrons on defects [35–37]. In these images, the spectral components are compressed in the vicinity of point Γ (Figure 4). The components of higher orders on Fourier images of the interlayer surface in solid solutions based on $Bi_2Te_3$ were found using scanning tunneling microscopy with atomic resolution [38].

The grain parameters were determined from the analysis of the surface morphology images of the p–$Bi_{0.5}Sb_{1.5}Te_3$ films obtained by discrete evaporation (Table 2). It was shown that the film deposited on a mica substrate that was annealed possesses a larger average area of grains than the unannealed ones deposited on a polyimide substrate. In addition, annealing nearly halves the number of grains from 69 to 37 in the studied films, accompanied by an increase in average grain areas, which is in good agreement with [16,17,39].

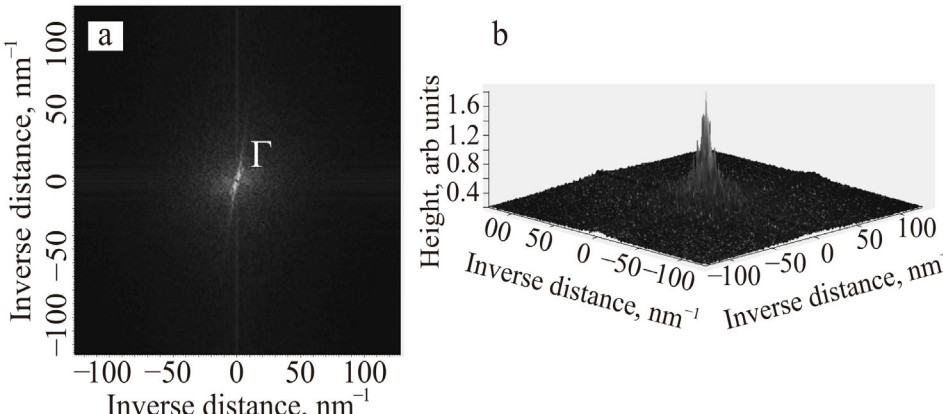

**Figure 4.** Fourier images of the surface morphology (0001) of the annealed p–$Bi_{0.5}Sb_{1.5}Te_3$ film obtained by the discrete evaporation method on a mica substrate. (**a**) Two-dimensional and (**b**) three-dimensional Fourier images, respectively.

**Table 2.** The average grain area <S> and areas of the grains of various sizes S1–S4 determined from the images of surface morphology (0001) in the unannealed (sample 1) and the annealed (sample 4) p–$Bi_{0.5}Sb_{1.5}Te_3$ films.

| No. in Table 1 | <S>, $\mu m^2$ | S1, $\mu m^2$, % | S2 $\mu m^2$, % | S3 $\mu m^2$, % | S4 $\mu m^2$, % |
|---|---|---|---|---|---|
| 1 | 0.135 | 0.001 45% | 0.002–0.008 43% | 0.01–0.05 10% | 2.6–8.7 2% |
| 4 | 1.625 | 0.002–0.077 72% | 0.15–0.92 18% | (9.5–41.25) 10% | |

## 4. Thermal Conductivity

The thermoelectric properties of the formed films were measured using the Physical Property Measurement System (PPMS) controlled by the Quantum Design Thermal Transport Option (TTO) software. The thermal conductivity of a material was measured by a TTO system as the temperature drops along the sample as a known quantity of heat that passes through the sample. The Seebeck coefficient was obtained by TTO as an electrical voltage drop caused by the temperature gradient across the studied sample. The electrical resistivity measurements were carried out by the PPMS system using the standard four-probe method. The PPMS allows for simultaneous measurements of thermoelectric properties over a wide temperature range from 2 to 390 K with a temperature change rate of $\pm 0.5$ K/min. The typical measurement accuracy of the Seebeck coefficient and thermal conductivity is $\pm 5\%$, and the typical precision of electrical resistivity is 0.01% for 1 $\Omega$ and 200 $\mu A$.

The total thermal conductivity of the p–$Bi_{0.5}Sb_{1.5}Te_3$ and p–$Bi_2Te_3$ films can be represented as: $\kappa = \kappa_L + \kappa_e$, where $\kappa_L$ and $\kappa_e$ are the crystal lattice and the electronic thermal conductivities, respectively. According to the Wiedemann–Franz law, the relationship between the electronic thermal conductivity $\kappa_e$ and electrical conductivity $\sigma$ is given by $\kappa_e = L(r, \eta)\sigma T$, where $L(r, \eta)$ is the Lorentz number:

$$L(r, \eta) = \left(\frac{k}{e}\right)^2 \left(\frac{(r + 7/2)F_{r+5/2}(\eta)}{(r + 3/2)F_{r+1/2}(\eta)} - \frac{(r + 5/2)^2 F_{r+3/2}^2(\eta)}{(r + 3/2)^2 F_{r+1/2}^2(\eta)}\right) \tag{1}$$

where k is the Boltzmann's constant, e is the charge of an electron, r is the scattering parameter, and $\eta$ is the reduced Fermi level. $F_{r+n}(\eta)$ is the Fermi function at n = 0.5, 1.5, and 2.5 in the form:

$$F_s(\eta) = \int_0^\infty \frac{x^s}{e^{x-\eta}+1}dx \tag{2}$$

where s = r + n.

The Lorentz number (1) was calculated for a single parabolic band with arbitrary degeneracy in the isotropic relaxation time approximation as a power law:

$$\tau = \tau_0 E^r \tag{3}$$

The constant $\tau_0$ does not depend on energy E, and r is the current value of the effective scattering parameter $r_{eff}$ [40]. The parameter $r_{eff}$ and the reduced Fermi level $\eta$ were determined by the least squares method from the experimental values of the thermoelectric power coefficient $\alpha(r, \eta)$ for the studied films and the degeneracy parameter $\beta_d(r, \eta)$ in accordance with (2–5). The latter was calculated within the framework of the many-valley model of the energy spectrum from the ratios of isotropic factors in the expressions for electrical conductivity, Hall conductivity, and magnetoconductivity as discussed in [40]. The expressions for the thermoelectric power coefficient $\alpha(r, \eta)$ and the degeneracy parameter $\beta_d(r, \eta)$ in the isotropic relaxation time approximation (3):

$$\alpha = \frac{k}{e}\left[\frac{(r+5/2)F_{r+3/2}(\eta)}{(r+3/2)F_{r+1/2}(\eta)} - \eta\right] \tag{4}$$

$$\beta_d(r,\eta) = \frac{(2r+3/2)^2 F^2_{2r+1/2}(\eta)}{(r+3/2)(3r+3/2)F_{r+1/2}(\eta)F_{3r+1/2}(\eta)} \tag{5}$$

The dependences $F_{r+n}(\eta)$, $r_{eff}(\eta)$, and $\alpha(\eta)$ (as shown in Figures 5–7, curve 7) illustrate the variation ranges of the reduced Fermi level $\eta$ and the $r_{eff}$ parameter corresponding to the experimental values of the thermopower coefficient $\alpha$ in the temperature range of 40–300 K. The obtained $r_{eff}$ parameter (Figure 6) differs from the value of r = −0.5, which is specific to the acoustic scattering mechanism. However, in bulk thermoelectrics, $|r_{eff}|$ is typically less than in the films [40].

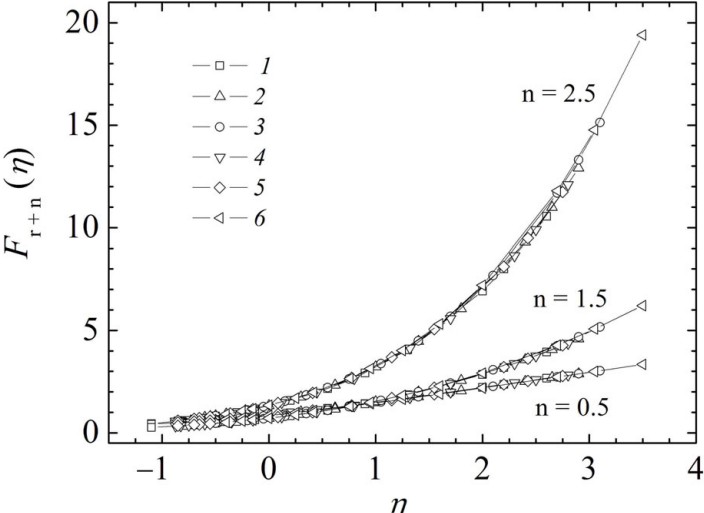

**Figure 5.** Fermi functions $F_{r+n}(\eta)$ for the p–Bi$_{0.5}$Sb$_{1.5}$Te$_3$ (1−4) and p–Bi$_2$Te$_3$ (5, 6) films. Sample numbers in figure and subsequent figures correspond to Table 1.

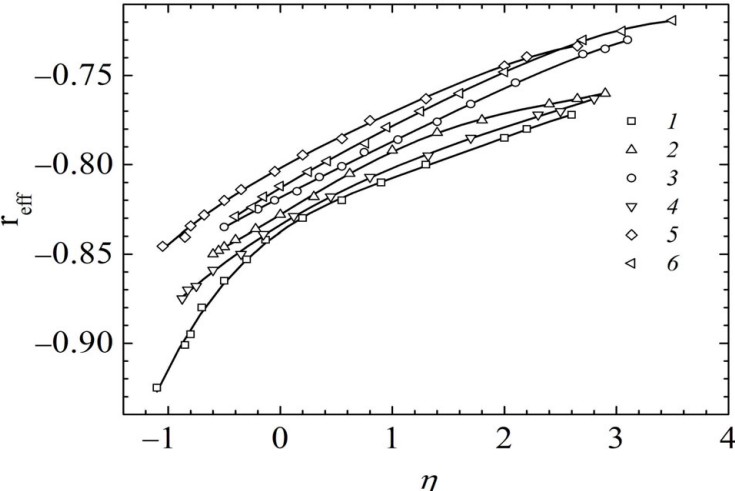

**Figure 6.** The dependences of the effective scattering parameter of the charge carriers ($r_{eff}$) on the reduced Fermi level $\eta$ in the p–Bi$_{0.5}$Sb$_{1.5}$Te$_3$ (1–4) and p–Bi$_2$Te$_3$ (5, 6) films.

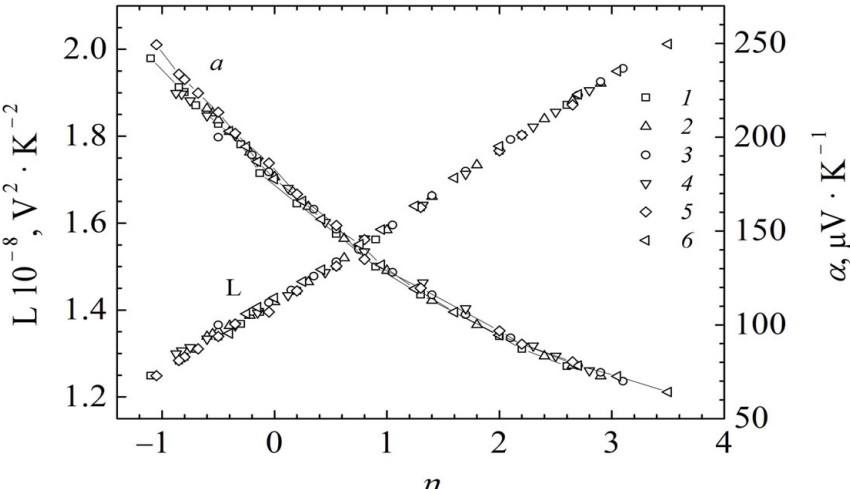

**Figure 7.** The dependences in the thermoelectric power coefficient $\alpha$ and the Lorentz number L on reduced Fermi level $\eta$ in the p–Bi$_{0.5}$Sb$_{1.5}$Te$_3$ (1–4) and p–Bi$_2$Te$_3$ (5, 6) films.

The increase in $|r_{eff}|$ observed in these films determines the enhancement in the relaxation time energy dependence (3) in the films of 3D-TIs [15,41,42]. The strong dependence of relaxation time on energy in TIs is related to the properties of the surface states of the Dirac fermions, which are protected by time-reversal symmetry from backscattering on non-magnetic defects. It promotes the creation of channels that are perfect for rapid, low-dissipative electrical transport and improvement in thermoelectric properties [15]. The mechanism of Dirac fermions scattering and the dependence of $\tau(E)$ in TIs have been studied in [41,42] within the framework of non-equilibrium Boltzmann transport theory. Estimations of $\tau(E)$ were carried out for cases of long-range Coulomb scattering (LCS) and short-range delta scattering (SDS). It was found that the dependence in the limit gives $\tau(E) \propto E$ for LCS and $\tau(E) \propto E^{-1}$ for SDS.

In the Dirac semimetal system, such as graphene with Fermi energy ($E_F$) located near the neutral Dirac point, the LCS dominates [41–43], and the relaxation time reaches the limit of $\tau(E) \propto E$. For 3D-TIs based on bismuth and antimony chalcogenides with low thermal conductivity, the limiting dependence of $\tau(E) \propto E^{-1}$ corresponds to the SDS on neutral impurity or phonon scattering [41,42].



The $r_{eff}$ values closest to the limiting $\tau(E) \propto E^{-1}$ were obtained in an unannealed film of p–$Bi_{0.5}Sb_{1.5}Te_3$ formed by discrete evaporation on a polyimide substrate (Figure 6). Here, the $r_{eff}$ varies from $-0.925$ to $-0.77$ depending on the reduced Fermi level $\eta$ (Figure 6, curve 1). According to scanning tunneling spectroscopy data [38], in the films of similar composition with an energy gap of 240 meV, the Fermi energy relative to the conduction band edge is $E_F = 105$ meV, and the Dirac point at $E_D = -130$ meV is located close to the valence band edge [15,41,42]. In this case, the dependence of $\tau(E)$ becomes close to the limiting $\tau(E) \propto E^{-1}$ [15,43]. The more noticeable difference in $r_{eff}$ from $-1$ in the p–$Bi_{0.5}Sb_{1.5}Te_3$ and p–$Bi_2Te_3$ films, as shown in Figure 6, is assumed to be caused by the potential fluctuations arising from bulk carriers [15,43].

The Lorentz number $L(r, \eta)$ (1) employed in the calculation of $\kappa_e$ was obtained using the data shown in Figures 5–7. The values of the Lorentz number in Figure 7 are consistent with [44] and are significantly lower than $L_0 = \pi^2 k^2/3e^2 = 2.44 \ 10^{-8}$ W$\Omega$K$^{-2}$ for complete degeneracy since the studied films partially degenerate at thermoelectric power $\alpha = 242–200$ µV K$^{-1}$ (see Table 1). According to estimations the use of L in calculations for the case of complete degeneracy can lead to errors of up to 40% [45].

Temperature dependences in the total thermal conductivity $\kappa$ in the films of p–$Bi_{0.5}Sb_{1.5}Te_3$ and p–$Bi_2Te_3$ were measured on samples with a thickness of about 1 µm. The effects associated with the surface states of the Dirac fermions can be observed not only in ultra-thin samples, owing to large quantum phase coherence length $l_\varphi$ related to the inelastic electron scattering processes. The value of $l_\varphi$ is usually significantly higher than the electron mean free path $l_F$, which allows the presence of the topological surface states to be revealed by transport property studies in both the nanometer-thick films [7,46–48] and also in submicron-thick films [15]. The obtained $r_{eff}(\eta)$ (Figure 6) demonstrates an enhancement in the energy dependence of the relaxation time (3) that confirms the possibility of investigation of TSS in submicron-thick films.

The temperature dependences of the total $\kappa$, lattice $\kappa_L$ and electron $\kappa_e$ thermal conductivities of the p–$Bi_{0.5}Sb_{1.5}Te_3$ and p–$Bi_2Te_3$ films shown in Figures 8–10 demonstrate that these values depend on the composition, film deposition method, and subsequent thermal treatment. It is known that in TIs, a tight coupling between spin and momentum prevents the backscattering of fermions on nonmagnetic impurities and defects. However, the residual bulk conductivity associated with the presence of bulk intrinsic defects [25,26] remains, which determines the scattering processes in the films under study.

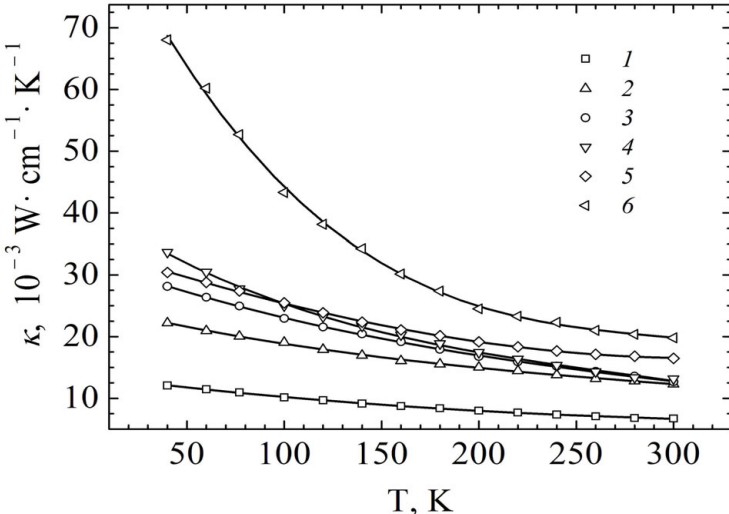

**Figure 8.** The temperature dependences of the total thermal conductivity $\kappa$ in the p–$Bi_{0.5}Sb_{1.5}Te_3$ (1–4) and p–$Bi_2Te_3$ (5, 6) films.

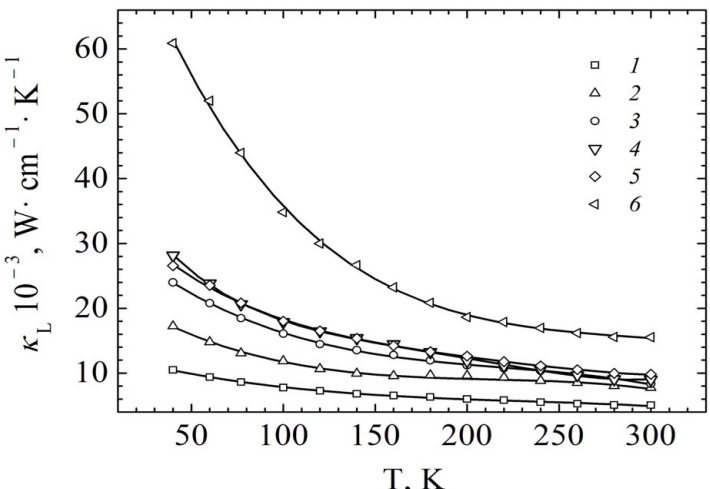

**Figure 9.** Temperature dependencies in the lattice thermal conductivity $\kappa_L$ in the p–Bi$_{0.5}$Sb$_{1.5}$Te$_3$ (1–4) and p–Bi$_2$Te$_3$ (5, 6) films.

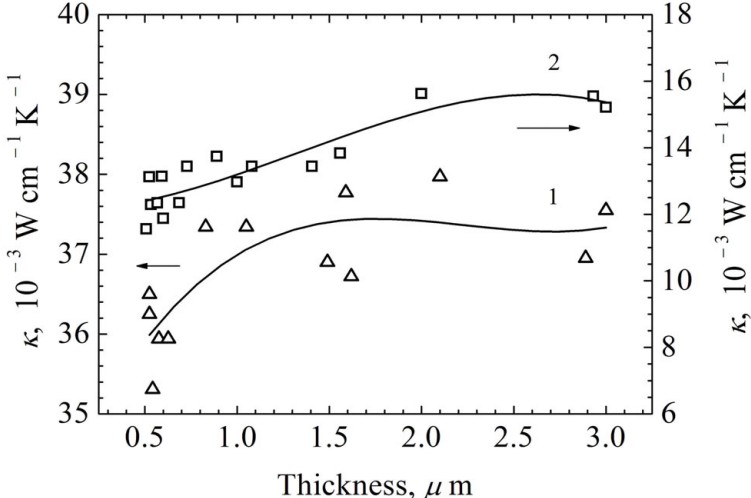

**Figure 10.** Dependences of the total thermal conductivity $\kappa$ on thickness in p–Bi$_2$Te$_3$ (1) and p–Bi$_{0.5}$Sb$_{1.5}$Te$_3$ (2) films obtained by discrete evaporation at T = 77 K.

The typical decrease in thermal conductivity in the films with an increase in temperature is consistent with the Peierls theory of three-phonon scattering processes, according to which $\kappa \propto T^{-1}$ at $T \geq T_D$ [49]. The weakening of the thermal conductivity temperature dependencies compared to $T^{-1}$ is determined by additional scattering processes, which are influenced by the film formation technique, film composition, and thickness.

It should be noted that calculations of the electronic thermal conductivity $\kappa_e$ were carried out, taking into account the parameter $r_{eff}(\eta)$ (Figure 6) [40]. A significant decrease in $\kappa$ and $\kappa_L$, accompanied by the weakening of temperature dependence of $\kappa(T)$ and $\kappa_L(T)$, was observed in the p–Bi$_{0.5}$Sb$_{1.5}$Te$_3$ solid solution (Figures 8 and 9, samples 1 and 2), deposited by discrete evaporation on a polyimide substrate. However, the largest decrease in $\kappa$ and $\kappa_L$ in the whole studied temperature range was obtained in the films that were not subjected to heat treatment (Figures 8 and 9, sample 1).

The values of $\kappa$ and $\kappa_L$ were increased and the dependence of $\kappa(T)$ and $\kappa_L(T)$ was enhanced in the low-temperature range in the p–Bi$_{0.5}$Sb$_{1.5}$Te$_3$ films, deposited with the discrete evaporation method on the mica substrates followed by annealing in the Ar atmosphere (Figures 8 and 9, sample 3). A further increase in $\kappa$ and $\kappa_L$ and rise in $\kappa(T)$ and $\kappa_L(T)$ were observed in the unannealed p–Bi$_{0.5}$Sb$_{1.5}$Te$_3$ films, formed by thermal

evaporation on the polyimide substrate (Figures 8 and 9, sample 4). In the unannealed p–$Bi_2Te_3$ films deposited on the polyimide substrate obtained by discrete evaporation, the values of $\kappa$ and $\kappa_L$ were significantly higher (Figures 9 and 10, sample 5) than in the p–$Bi_{0.5}Sb_{1.5}Te_3$ solid solution (Figures 8 and 9, sample 1). The largest values of $\kappa$ and $\kappa_L$ in the p–$Bi_2Te_3$ films were obtained when thermal evaporation was used (Figures 8 and 9, sample 6). The total $\kappa$ and lattice $\kappa_L$ thermal conductivity values (Figures 8 and 9) were in good agreement with the data for the $Bi_2Te_3$ films obtained by mechanical exfoliation [39], for the films deposited onto the polyimide substrates by thermal evaporation [50], and for nanocrystalline composite p–$Bi_{0.52}Sb_{1.48}Te_3$ solid solutions [17].

The considered dependences of $\kappa$(T) and $\kappa_L$(T) show that the optimization of technological parameters affects the intensity of the scattering of long-wavelength phonons on the grain interfaces in the films. Thus, the optimization allows for a significant reduction in the values of $\kappa$ and $\kappa_L$ at temperatures above the Debye temperature of $T_D$ in the unannealed p–$Bi_{0.5}Sb_{1.5}Te_3$ films, obtained by discrete evaporation on the polyimide substrate [16,22–24]. The decrease in $\kappa$ and $\kappa_L$ and the weakening of the $\kappa$(T) and $\kappa_L$(T) at temperatures below 100 K is determined by the scattering of phonons on intrinsic antisite defects of $Bi_{Te}$ and impurity defects that originated during the formation of the p–$Bi_{0.5}Sb_{1.5}Te_3$ films [19,20]. Additionally, in these films with low values of $\kappa$ and $\kappa_L$, the thermoelectric power coefficient $\alpha$ increases (Figures 8 and 9, samples 1 and 4, Table 1) due to the effect of energy filtering [14,15].

As shown in Figures 1, 3, 8 and 9, a correlation was established between the thermal conductivity in the p–$Bi_{0.5}Sb_{1.5}Te_3$ films obtained by discrete evaporation and the images of surface morphology (0001). In the unannealed p–$Bi_{0.5}Sb_{1.5}Te_3$ films (Figures 8 and 9, curve 1), the average heights $R_a$, the root mean square deviations of the heights $R_q$ of nanofragments, and the area and number of grains on the (0001) surface morphology images are increased. Meanwhile, the values of the thermal conductivities of $\kappa$ and $\kappa_L$ are reduced compared to the annealed films (Figures 8 and 9, curves 2 and 4).

Moreover, studies on the total thermal conductivity $\kappa$ versus thickness were carried out on the p–$Bi_2Te_3$ and p–$Bi_{0.5}Sb_{1.5}Te_3$ films obtained by discrete evaporation. It could be considered that at the film thickness of 0.5 µm a tendency towards a decrease in thermal conductivity is revealed in some samples of both the p–$Bi_2Te_3$ and the p–$Bi_{0.5}Sb_{1.5}Te_3$ films (Figure 10). A significant decrease in the thermal conductivity $\kappa$ to 13 and 25 $10^{-3}$ W $cm^{-1}$ $K^{-1}$ was obtained in [21] at 100 K in ultra-thin p–$Bi_2Te_3$ films with a thickness of 3.03 and 9.09 nm, respectively.

The dependence of thermal conductivity on the thickness of p–$Bi_{0.5}Sb_{1.5}Te_3$ films (Figure 10) associated with the influence of Dirac fermions surface states is consistent with micro-Raman spectroscopy studies [51]. The appearance of inactive longitudinal phonons $A_{1u}{}^2$ in the Raman spectra of p–$Bi_{0.5}Sb_{1.5}Te_3$ films becomes noticeable at a film thickness of about 500 nm. The intensity of the $A_{1u}{}^2$ mode and the intensity ratios of the inactive longitudinal $A_{1u}{}^2$ and active transverse $E_g{}^2$ optical phonon modes are found to increase with decreasing thickness [51]. Such a dependence of the intensities of longitudinal and transverse modes on the film thickness is determined by the distortion of the inversion symmetry of the crystal, and characterizes the effects associated with electronic surface states of Dirac fermions [52,53]. An inactive $A_{1u}{}^2$ mode of weak intensity was also observed in micro-Raman spectra of $Bi_2Te_3$ measured at pressures of about 3 GP [54], at which a topological surface phase transition was observed [55].

The properties of the topological surface states of Dirac fermions in micron-thick Sn$-$(Bi,Sb)$_2$(Te,S)$_3$ films with a tetradymite structure were investigated in thicker films [15,56]. The estimations of the ratios of surface conductivity $G_s$ to the total conductivity $G_{tot}$ in the films showed that $G_s/G_{tot}$ = 80% for film thicknesses of 3 µm at temperatures up to 210 K [15] and for 1 µm thick films at T up to 125 K [56], which confirms the domination of the surface transport properties.

In addition to the dependences of the total thermal conductivity $\kappa$ on thickness in p–$Bi_2Te_3$ and p–$Bi_{0.5}Sb_{1.5}Te_3$, similar dependences of resistivity $\rho$ were considered for

$Bi_2Te_3$ films grown by the molecular-beam epitaxy (MBE) method [25]. Table 3 shows that the values of $\rho$ decreased with increasing thickness.

**Table 3.** Dependence of resistivity $\rho$ and electrical conductivity $\sigma$ on $Bi_2Te_3$ film thickness.

| Sample | T, K | t | $\rho$, μΩ m | $\sigma$, $\Omega^{-1}cm^{-1}$ |
|--------|------|-----|--------------|-------------------------------|
| [25] | 100 | 30 nm | 5.33 | 1877 |
| [25] | 100 | 16 nm | 3.11 | 3217 |
| [25] | 20 | 30 nm | 4.44 | 2252 |
| [25] | 20 | 16 nm | 1.776 | 5630 |
| 5 | 100 | 1 μm | 2.33 | 4275 |
| 6 | 100 | 1 μm | 2.2 | 4532 |
| 5 | 20 | 1 μm | 0.98 | 10230 |
| 6 | 20 | 1 μm | 1.12 | 8930 |

Samples 5 and 6 of the data in Table 3 are taken from Figure 11, inset.

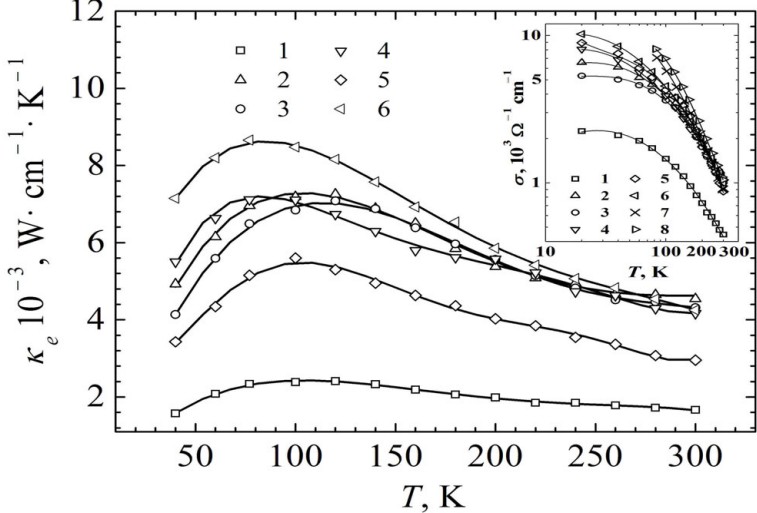

**Figure 11.** The temperature dependencies of the electronic thermal conductivity $\kappa_e$ and the electrical conductivity $\sigma$ (inset) in the p–$Bi_{0.5}Sb_{1.5}Te_3$ (1–4) and p–$Bi_2Te_3$ (5, 6) films. The designations of the samples (1–6) correspond to those in the Table 1, and 7 and 8 are the bulk samples of the p–$Bi_{0.5}Sb_{1.5}Te_3$ solid solution. In the films of p–$Bi_{0.5}Sb_{1.5}Te_3$ (1–3, inset), the angular coefficients of $d\ln\sigma/d\ln T = 0.1$ in the temperature interval T = (40–80) K and 1–1.2 in the interval T = (100–300) K. Resistivity ratios $\rho(300\ K)/\rho(20\ K)$ were obtained as $\rho = 1/\sigma$ from the inset: 1–5, 2–6.1, 3–5.3, 4–7.9, 5–10.2, 6–10.2.

In the submicron p–$Bi_2Te_3$ films studied in this work, the resistivity was significantly lower, and the electrical conductivity was, accordingly, higher [25] (Table 3). This was achieved by optimizing the film formation technological parameters, which resulted in a decrease in bulk point defects accompanied by an increase in electrical conductivity due to an enhancement in the contribution of surface electrical conductivity. Thus, such thickness-dependent analysis of $\rho$ should reasonably be made for films obtained under similar growth conditions.

Temperature dependences of the electronic thermal conductivity $\kappa_e$ are determined by the corresponding electrical conductivity $\sigma(T)$ (Figure 11, inset), which includes the contribution of both the metallic surface and the bulk electrical conductivity of TIs films [25–27]. The variation in the technological parameters of the film formation allows us to reduce the bulk conductivity by optimizing the scattering of electrons on intrinsic antisite defects of $Bi_{Te}$ and impurity substitution defects of Sb → Bi atoms in the p–$Bi_{0.5}Sb_{1.5}Te_3$ solid solution.

In the low-temperature range, an enhancement in $\kappa_e$ with an increase in the temperature from 40 to 80 K was observed (Figure 11, curves 1–3, inset), and a significant decrease in the angular coefficients of $\mathrm{dln}\sigma/\mathrm{dlnT}$ was obtained compared to the interval of 100–300 K in the p–$Bi_{0.5}Sb_{1.5}Te_3$ films. The revealed feature of $\kappa_e(T)$ at low temperatures is explained by the enhancement in the energy dependence of relaxation time $\tau$ due to the rise in $|r_{eff}|$, which was used in the Lorentz number (1) calculations.

The increase in $|r_{eff}|$ with decreasing Fermi level $\eta$ (Figure 6) is associated with the strengthening of electron scattering on intrinsic and impurity substitution defects at low temperatures [19,20]. Intensive scattering on defects with a temperature decrease from 80 to 40 K determines the reduction in electrical conductivity $\sigma$ and corresponding decrease in electronic thermal conductivity $\kappa_e$. The "hump" on the $\kappa_e(T)$ dependence observed in the temperature range of 80–110 K can be explained by potential fluctuations due to the influence of bulk carriers. These fluctuations result in the deviation of the energy dependence of relaxation time from the limiting $\tau(E) \propto E^{-1}$ [15,41,42], and therefore to the decrease in the $|r_{eff}|$ parameter with rise in $\eta$ (Figure 6).

The temperature dependences $\sigma(T)$ (Figure 11, inset) were analyzed using the $\rho = 1/\sigma$, obtained as resistivity ratios between values of $\rho$ measured at room (300 K) and low (20 K) temperatures in accordance with [29,57]. The observed decrease in the slopes $\sigma(T)$ at low temperatures down to 80 K (Figure 11, inset) turned out to be insufficient to change the $\sigma(T)$ of the films from a metallic conductivity type to a semiconductor one, in contrast to [27], where the ratios of the $\rho(10\ K)/\rho(300\ K)$ were considered.

Our studies have shown that the ratios of $\rho(300\ K)/\rho(20\ K)$ (Figure 11) as well as the total $\kappa$ and the lattice $\kappa_L$ thermal conductivities (Figures 8 and 9) obtained in the p–$Bi_{0.5}Sb_{1.5}Te_3$ are lower than in the p–$Bi_2Te_3$ regardless of the film formation technique used. A decrease in the ratios $\rho(300\ K)/\rho(20\ K)$ in the p–$Bi_{0.5}Sb_{1.5}Te_3$ films obtained by discrete evaporation is accompanied by a decrease in $\kappa$ and $\kappa_L$ (Figures 8 and 9, samples 1, 2, and 4).

The carried-out estimations of the thermoelectric figure of merit ZT, where $Z = \alpha^2\sigma/\kappa$, showed that in the p–$Bi_{0.5}Sb_{1.5}Te_3$ films formed by discrete evaporation, the highest ZT values near room temperature were 1, 1.12, and 1.08 in samples 1, 2, and 4 (Table 1), respectively. In the p–$Bi_{0.5}Sb_{1.5}Te_3$ films (samples 1 and 2) deposited on a polyimide substrate, the ratios of $\rho(300\ K)/\rho(20\ K)$ were lower than in sample 4 obtained on a mica substrate, which possessed high electrical conductivity (Figure 11, curve 4, inset). In the p–$Bi_2Te_3$ films (samples 5 and 6, Table 1) with high ratios $\rho(300\ K)/\rho(20\ K)$ of about 10, the ZT values decreased from 0.87 upon discrete evaporation to 0.65 upon thermal evaporation. In the bulk samples of the p–$Bi_{0.5}Sb_{1.5}Te_3$ solid solution (Figure 11, curves 7 and 8, inset), the ratio is about 8. Thus, the resistivity ratios are $\rho(300\ K)/\rho(20\ K)$ and can be used to analyze the thermoelectric properties of topological insulator films based on p–$Bi_2Te_3$.

It is well known from recent studies that the values of the thermoelectric figure of merit in chalcogenide films of TIs can vary over a wide range depending on the formation technique and composition. Therefore, near room temperature, the values of ZT were obtained as 0.9 in the p–$Bi_2Te_3$ [50] and the p–$Bi_{0.5}Sb_{1.5}Te_3$ [57] films, 0.37 in p–$Bi_{0.5}Sb_{1.5}Te_3$ films [58], and 1.44 in the p–type $Bi_2Te_3/Bi_{0.5}Sb_{1.5}Te_3$ superlattice films with a low interfacial resistance [59].

More detailed information about the properties of the surface states of Dirac fermions was obtained from studies on quantum oscillations of the magnetoresistance in strong magnetic fields [60–62], angle-resolved photoemission spectroscopy [63,64], and differential tunneling conductance spectra using scanning tunneling spectroscopy [35,38,65,66]. The existence of the surface states of Dirac fermions in the p–$Bi_{0.5}Sb_{1.5}Te_3$ and the p–$Bi_2Te_3$ submicron films was confirmed from studies of differential tunneling conductance spectra by scanning tunneling spectroscopy at room temperature [38]. In the studied p–$Bi_{0.5}Sb_{1.5}Te_3$ and p–$Bi_2Te_3$ films, the values of the thermoelectric power coefficient are $\alpha = 178\ \mu V\ K^{-1}$ and $272\ \mu V\ K^{-1}$, and the surface concentration of fermions $n_s$ is $4\ 10^{12}\ cm^{-2}$ and $1.2\ 10^{13}\ cm^{-2}$, respectively. The quantum Hall effect, the parameters of the surface

states of Dirac fermions, and the metallic conductivity on the surface of the p–$Bi_{0.5}Sb_{1.5}Te_3$ and the p–$Bi_2Te_3$ films, obtained from studies of galvanomagnetic properties in strong magnetic fields, confirm the topological properties of submicron-thick films [15,56] and the prospects for practical application.

Currently, high-pressure methods are employed for the optimization of thermoelectric properties. In the developed p–$Bi_{0.5}Sb_{1.5}Te_3$ films, considered in this study, topological phase transitions (TPTs) were found at pressures of P = 3–4 GPa as a result of pressure-dependent thermoelectric property investigations [34]. Near the TPTs at room temperature, the power factor increases by more than two times, while the thermal conductivity rises slightly [67], which makes the considered films promising as the p-branch for the development of thermoelectric modules [68]. The module consists of a specially designed high-pressure cell compressed between two opposing anvils.

Multicomponent solid solutions $Bi_2(Te, Se, S)_3$ were used as the n-branch of thermoelements. The performance parameters of the high-pressure module can be optimized by changing the pressure applied to its elements [68]. Additionally, the thermoelectric parameters can be enhanced due to the appearance of additional transverse and longitudinal magneto-thermoelectric effects in an external magnetic field [69].

## 5. Conclusions

Investigations of the thermal conductivity of the p–$Bi_{0.5}Sb_{1.5}Te_3$ and p–$Bi_2Te_3$ films of TIs formed by discrete and thermal evaporation techniques were carried out. The study revealed that the total $\kappa$, lattice $\kappa_L$, and electronic $\kappa_e$ thermal conductivities depend not only on the composition and optimization of the film formation parameters but also on the influence of the energy dependence of the relaxation time. Additionally, the effective scattering parameter of $r_{eff}$ was taken into account while calculating the $\kappa_e$ values using the Lorentz number L(r, $\eta$). The unannealed films of the p–$Bi_{0.5}Sb_{1.5}Te_3$ solid solution formed by discrete evaporation on polyimide substrates exhibited a significant decrease in $\kappa$, $\kappa_L$, and $\kappa_e$ values.

The reduction in the crystal lattice thermal conductivity $\kappa_L$ in TI films is associated with the enhancement in the energy dependence of the mean free path of phonons, which leads to the intensive scattering of long-wavelength phonons on the grain interfaces. At temperatures near and above $T_D$ up to room temperature, the contribution of phonon–phonon scattering increases. In the range of low temperatures, the main reason for the decrease in $\kappa_L$ is the scattering of short-wavelength phonons on intrinsic point defects and impurities. Furthermore, an additional decrease in $\kappa_L$ also occurs due to the scattering of phonons on the interfaces of the Te(1) layers of the van der Waals gap.

The reduction in $\kappa_e$ with temperature is explained by a decrease in the electrical conductivity $\sigma$ in the films caused by electron scattering on the intrinsic antisite $Bi_{Te}$ and impurity substitution defects of Sb → Bi. The peculiarity of the dependence $\kappa_e$(T) in the temperature range of 80–110 K in the form of a hump is explained by the influence of potential fluctuations in bulk carriers. These fluctuations lead to a decrease in the effective scattering parameter $|r_{eff}|$ with increasing Fermi level due to the deviation of the energy dependence of relaxation time from the limiting $\tau(E) \propto E^{-1}$.

The study also investigated the effect of heat treatment on the relief properties of the interlayer surface (0001) studied by AFM and on the variation in the thermal conductivity in the p–$Bi_{0.5}Sb_{1.5}Te_3$ films. The results show that the total $\kappa$ and the crystal lattice $\kappa_L$ thermal conductivities are reduced, but the number of grains on the surface (0001) and the nanofragment parameters (the average heights $R_a$ and the root mean square deviations in the heights $R_q$) are increased in the unannealed films compared to the annealed ones.

Finally, the low thermal conductivity and high electrical conductivity obtained in annealed p–$Bi_{0.5}Sb_{1.5}Te_3$ films formed by discrete evaporation on a polyimide substrate provide an increase in the thermoelectric figure of merit. The observed increase in ZT up to 1.12 near room temperature is associated with an enhancement in the energy dependence of the relaxation time in TIs caused by an increase in the effective scattering parameter $r_{eff}$.

**Author Contributions:** Conceptualization, L.N.L.; Data curation, L.N.L., O.A.U. and Y.A.B.; Formal analysis, L.N.L. and O.A.U.; Investigation, Y.A.B. and I.V.M.; Methodology, V.A.D., I.V.M. and V.N.P.; Resources, L.N.L., Y.A.B., V.A.D. and I.V.M.; Software, O.A.U.; Writing—original draft, L.N.L. and O.A.U.; Writing—review & editing, L.N.L. All authors have read and agreed to the published version of the manuscript.

**Funding:** This research was funded by Russian Foundation for Basic Research grant number 20-08-00464.

**Data Availability Statement:** The data that support the findings of this study are available from the corresponding author upon reasonable request.

**Conflicts of Interest:** The authors declare no conflict of interest.

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
