# Peer review of "Thermal Conductivity for p–(Bi, Sb)2Te3 Films of Topological Insulators"

_magnetochemistry, doi:10.3390/magnetochemistry9060141_

Round 1

Reviewer 1 Report

The refereed manuscript is devoted to study of the thermal conductivity in films of topological insulators p–Bi0.5Sb1.5Te3 and p–Bi2Te3. The authors state that “in the p– Bi0.5Sb1.5Te3 films with low thermal conductivity the energy dependences of the re-laxation time are enhanced, which is specific to the topological insulators. The electronic thermal conductivity was determined by taking into account the effec-tive scattering parameter in the relaxation time approximation versus energy in the Lo-rentz number calculations.” The manuscript is interesting and with potential for publication. However, some issues have to be attended before the acceptance of the manuscript.

1- The discussion of the results suffers from the physics of the problem. The authors essentially limit themselves to describing the captions of the figures in an enlarged way or to illustrate what is observed in them without entering into justifying or giving arguments that support these results. 

2- It will be better understanding and interesting for reading the manuscript if some practical application (example) of the study is presented in the manuscript.

3- The references are not enough (for example, reference is required after the ”Currently, bismuth and antimony chalcogenides attract much attention as promising topological insulators (TIs), in which topological surface states (TSS) arise due to inversion of the energy gap edges caused by strong spin-orbit inter-action.).

4- The English language of the paper must strongly be checked with the help of a native speaker.

5- There are many figures and the explanation of some figures is very brief, and the quality of the some figures is not high enough (such as Figs. 1(a) and 1(b), 2 and inset of Fig. 11).

6- The manuscript should be written with the same font.

7- I suggest that authors plot the schematic of the investigated device.

8- The Raman spectra of the films are needed.

Author Response

First of all, we would like to thank the Editor and the Reviewer for their constructive and insightful comments and suggestions to our manuscript. We modified the manuscript entitled “Thermal conductivity for p-(Bi, Sb)2Te3 films of topological insulators” as recommended. The modifications are summarized below.

Answers to Reviewer 1

Q1: The discussion of the results suffers from the physics of the problem. The authors essentially limit themselves to describing the captions of the figures in an enlarged way or to illustrate what is observed in them without entering into justifying or giving arguments that support these results. 

We have added detailed explanation to illustrate the obtained results in the figures 6, 10, 11. Explanation for figure 6:

(Page 9-10)

The strong dependence of relaxation time on energy in TIs is related to the properties of the surface states of the Dirac fermions, which are protected by time-reversal symmetry from backscattering on non-magnetic defects. It leads to the formation of channels that are perfect for fast and ultralow dissipative electrical transport and improvement of thermoelectric properties [15]. The mechanism of Dirac fermions scattering and the dependence of τ(E) in TIs have been studied in [40,41] within the framework of non-equilibrium Boltzmann transport theory. Calculations of τ(E) were carried out for cases of long-range Coulomb scattering (LCS) and short-range delta-scattering (SDS), which in the limit gives dependences of τ(E) µ E and τ(E) µ E-1, respectively.

In the Dirac semimetal system such as graphene with the Fermi energy (EF) being located near the neutral Dirac point, the LCS dominates [40–42], and the relaxation time reaches the limit of τ(E) µ E. For 3D-TIs based on bismuth and antimony chalcogenides with low thermal conductivity, the limiting dependence of τ(E) µ E-1 corresponds to the SDS on neutral impurity or phonon scattering [40,41].

The reff values closest to the limiting τ(E) µ E-1 were obtained in an unannealed film of p-Bi0.5Sb1.5Te3 formed by discrete evaporation on a polyimide substrate (figure 6). Here, the reff varies from -0.925 to -0.77 depending on the reduced Fermi level η (figure 6, curve 1). According to scanning tunneling spectroscopy data [37], in the films of similar composition with energy gap of 240 meV, the Fermi energy relative to the conduction band edge is EF = 105 meV, and the Dirac point at ED =-130 meV is located close to the valence band edge [15,40,41]. In this case, the dependence of τ(E) becomes close to the limiting τ(E) µ E-1 [15,42]. The more noticeable difference of reff from -1 in the p–Bi0.5Sb1.5Te3 and p–Bi2Te3 films, as shown in figure 6, is assumed to be caused by the potential fluctuations arising from bulk carriers [15,42].

[15] Matsushita, S.Y.; Ichimura, K.; Huynh, K.K.; Tanigaki, K. Large thermopower in topological surface state of Sn-BSTS topological insulators: Thermoelectrics and energy-dependent relaxation times. Phys. Rev. Mater. 2021, 5, 014205–(8pp). https://doi.org/10.1103/PhysRevMaterials.5.014205

[40] Nomura, K.; MacDonald, A.H. Quantum Transport of Massless Dirac Fermions. Phys. Rev. Lett. 2007, 98, 076602-(4pp). https://doi.org/10.1103/PhysRevLett.98.076602

[41] Nomura, K.; Koshino, M.; Shinsei, R. Topological Delocalization of Two-Dimensional Massless Dirac Fermions. Phys. Rev. Lett. 2007, 99, 146806-(4pp).

https://doi.org/10.1103/PhysRevLett.99.146806

[42] Chiba, T.; Takahashi, S. Transport properties on an ionically disordered surface of topological insulators: Toward high-performance thermoelectrics. J. Appl. Phys. 2019, 126, 245704-(13pp). (2019). https://doi.org/10.1063/1.5131311

Explanation for figure 10:

(Page 13)

The dependence of thermal conductivity on the thickness of p-Bi0.5Sb1.5Te3 films (figure 10) associated with the influence of Dirac fermions surface states is consistent with micro-Raman spectroscopy studies [49]. The appearance of inactive longitudinal phonons A1u2 in the Raman spectra of p-Bi0.5Sb1.5Te3 films becomes noticeable at a film thickness of about 500 nm. The intensity of the A1u2 mode and the intensity ratios of the inactive longitudinal A1u2 and active transverse Eg2 optical phonon modes are found to increase with decreasing thickness [49]. Such a dependence of the intensities of longitudinal and transverse modes on the film thickness is determined by the distortion of the inversion symmetry of the crystal, and characterizes the effects associated with electronic surface states of Dirac fermions [50,51]. An inactive A1u2 mode of weak intensity was also observed in micro-Raman spectra of Bi2Te3 measured at pressures of about 3 GP [52], at which a topological surface phase transition was observed [53].

The properties of topological surface states of Dirac fermions in micron-thick Sn-(Bi,Sb)2(Te,S)3 films with tetradymite structure were investigated in thicker films [15, 54]. The estimations of the ratios of surface conductivity Gs to the total conductivity Gtot in the films showed that Gs/Gtot= 80% for film thicknesses of 3 µm at temperatures up to 210 K [15] and for 1 µm thick films at T up to 125 K [54], which confirms domination of surface transport properties.

[49] Lukyanova, L.N.; Bibik, A.Yu.; Aseev, V.A.; Usov, O.A.; Makarenko, I.V.; Petrov, V.N.; Nikonorov, N.V.; Kutasov, V.A. Surface morphology and Raman spectroscopy of thin layers of antimony and bismuth chalcogenides. Phys. Solid State. 2016, 58, 1440–1447. https://doi.org/10.1134/S1063783416070258

[50] Teweldebrhan, D.; Goyal, V.; Balandin, A.A. Exfoliation and characterization of bismuth telluride atomic quintuples and quasi-two-dimensional crystals. Nano Lett. 2010, 10, 1209-(18pp). https://doi.org/10.1021/nl903590b

[51] Plucinski, L.; Herdt, A.; Fahrendorf, S.; Bihlmayer, G.; Mussler, G.; Doring, S.; Kampmeier, J.; Matthes, F.; Burgler, D. E.; Grutzmacher, D.; Blugel, S.; Schneider, C.M. J. Appl. Phys. 2013, 113, 053706-(6pp). https://doi.org/10.1063/1.4789353

[52] Ovsyannikov, S.V.; Morozova, N.V.; Korobeinikov, I.V.; et al. Enhanced power factor and high-pressure effects in (Bi,Sb)2(Te,Se)3 thermoelectrics Appl. Phys. Lett. 2015, 106, 143901–5. https://doi.org/10.1063/1.4916947

[53] Korobeinikov, I.V.; Morozova, N.V.; Lukyanova, L.N.; Usov, O.A.; Ovsyannikov, S.V. On the Power Factor of Bismuth-Telluride-Based Alloys near Topological Phase Transitions at High Pressures. Semiconductors. 2019, 53, 732–736. https://doi.org/10.1134/S1063782619060083

[54] Xu, Y.; Miotkowski, I.; Liu, C.; Tian, J.; Nam, H.; Alidoust, N.; Hu, J.; Shih, C.-K.; Hasan Z.; Chen, Y. P. Observation of topological surface state quantum Hall effect in an intrinsic three-dimensional topological insulator. Nat. Phys. 2014, 10, 956–963. https://doi.org/10.1038/nphys3140

Explanation for figure 11:

(Page 14-15)

In the low temperature range, an enhancement of κe with increase in the temperature from 40 to 80 K was observed (figure 11, curves 1–3, inset), and a significant decrease in the angular coefficients of dlnσ/dlnT was obtained compared to the interval of 100-300 K in the p–Bi0.5Sb1.5Te3 films. Hence, the revealed feature of κe(T) at low temperatures is explained by the enhancement of the energy dependence of relaxation time τ due to the rise of |reff|, which was used in the Lorentz number (1) calculations. The increase of |reff| with decreasing of Fermi level η (figure 6) is associated with the strengthening of electron scattering on intrinsic and impurity substitution defects at low temperatures [19,20]. Intensive scattering on defects with a temperature decrease from 80 to 40 K determines the reduction of electrical conductivity σ and corresponding decrease of electronic thermal conductivity κe. The "hump" on the κe(T) dependence observed in the temperature range of 80–110 K can be explained by potential fluctuations due to the influence of bulk carriers. These fluctuations result in deviation of the energy dependence of relaxation time from the limiting τ(E) µE-1 [15,40,41], and therefore to the decrease of the |reff| parameter with rise of η (figure 6).

Q2: It will be better understanding and interesting for reading the manuscript if some practical application (example) of the study is presented in the manuscript.

and Q7- I suggest that authors plot the schematic of the investigated device.

(Page 16)

Currently, high-pressure methods are employed for optimization of the thermoelectric properties. In the developed p-Bi0.5Sb1.5Te3 films, considered in this study, topological phase transitions (TPTs) were found at pressures of P = 3-4 GPa as a result of pressure-dependent thermoelectric property investigations [53]. Near the TPTs at room temperature, the power factor increases by more than two times, while the thermal conductivity slightly rises [59], which makes the considered films promising as the p-branch for development of thermoelectric modules [60]. The thermoelectric module model includes a miniature high-pressure cell and a specially designed thermoelectric module that is compressed between two opposing anvils. Multicomponent solid solutions Bi2(Te, Se, S)3 were used as the n-branch of thermoelements. The performance parameters of the high-pressure module can be optimized by changing the pressure applied to its elements [60]. Additionally, the thermoelectric parameters can be enhanced due to the appearance of additional transverse and longitudinal magneto-thermoelectric effects in an external magnetic field [61].

The schematic of the investigated device was published earlier in [68].

[53] Korobeinikov, I.V.; Morozova, N.V.; Lukyanova, L.N.; Usov, O.A.; Ovsyannikov, S.V. On the Power Factor of Bismuth-Telluride-Based Alloys near Topological Phase Transitions at High Pressures. Semiconductors. 2019, 53, 732–736. https://doi.org/10.1134/S1063782619060083

[67] M. K. Jacobsen, S. V. Sinogeikin, R. S. Kumar, and A. L. Cornelius, J. Phys. Chem. Solids 73, 1154-(5pp). (2012). https://doi.org/10.1016/j.jpcs.2012.05.001

[68] Korobeinikov, I.V.; Morozova, N.V.; Lukyanova, L.N.; Usov, O.A.; Kulbachinskii, V.A.; Shchennikov, V.V.; Ovsyannikov, S.V. Stress-controlled thermoelectric module for energy harvesting and its application for the significant enhancement of the power factor of Bi2Te3-based thermoelectrics. J. Phys. D: Appl. Phys. 2018, 51, 025501–(13pp). https://doi.org/10.1088/1361-6463/aa9b5f

[69] Morozova, N.V.; Korobeinikov, I.V.; Ovsyannikov, S.V. Strategies and challenges of high-pressure methods applied to thermoelectric materials. J. Appl. Phys. 2019, 125, 220901-(19pp). https://doi.org/10.1063/1.5094166

Q3: The references are not enough (for example, reference is required after the ”Currently, bismuth and antimony chalcogenides attract much attention as promising topological insulators (TIs), in which topological surface states (TSS) arise due to inversion of the energy gap edges caused by strong spin-orbit inter-action.”).

(Page 1)

Currently, bismuth and antimony chalcogenides attract much attention as promising topological insulators (TIs), in which topological surface states (TSS) arise due to inversion of the energy gap edges caused by strong spin-orbit interaction [3-5].

[3] Gilbert, M.J. Topological electronics. Comm. Phys. 2021, 4, 70–(12pp). https://doi.org/10.1038/s42005-021-00569-5

[4] Heremans, J.; Cava, R.; Samarth, N. Tetradymites as thermoelectrics and topological insulators. Nature Rev. Mater. 2017, 2 17049–(21pp). https://doi.org/10.1038/natrevmats.2017.49

[5] Ngabonziza, P. Quantum transport and potential of topological states for thermoelectricity in Bi2Te3 thin films. Nanotechnology. 2022, 33, 192001–(22pp). https://doi.org/10.1088/1361-6528/ac4f17

Q4: The English language of the paper must strongly be checked with the help of a native speaker.

English has been checked and improved throughout the manuscript. Some of long and complex sentences were improved for better reading.

Page 2, paragraph 2, 4. Page 3, paragraph 2. Page 8, paragraph 2. Page 12, paragraph 2, 3.

Page 13, paragraph 3. Page 16, paragraph 2, 4 (conclusion).

Q5:- There are many figures and the explanation of some figures is very brief, and the quality of the some figures is not high enough (such as Figs. 1(a) and 1(b), 2 and inset of Fig. 11).

The quality of figures 1 (a) and 1 (b), 2 and 11 have been improved.

Q6: The manuscript should be written with the same font.

The formatting has been checked to improve text appearance.

Q7: I suggest that authors plot the schematic of the investigated device.

Published earlier in [60], the schematic of the investigated device is described in Q2.

Q8: The Raman spectra of the films are needed.

(Page 13)

      The dependence of thermal conductivity on the thickness of p-Bi0.5Sb1.5Te3 films (Figure 10) associated with the influence of Dirac fermions surface states is consistent with micro-Raman spectroscopy studies [49]. The appearance of inactive longitudinal phonons A1u2 in the Raman spectra of p-Bi0.5Sb1.5Te3 films becomes noticeable at a film thickness of about 500 nm. The intensity of the A1u2 mode and the intensity ratios of the inactive longitudinal A1u2 and active transverse Eg2 optical phonon modes are found to increase with decreasing thickness [49]. Such a dependence of the intensities of longitudinal and transverse modes on the film thickness is determined by the distortion of the inversion symmetry of the crystal, and characterizes the effects associated with electronic surface states of Dirac fermions [50,51]. An inactive A1u2 mode of weak intensity was also observed in micro-Raman spectra of Bi2Te3 measured at pressures of about 3 GP [52], at which a topological surface phase transition was observed [53].

Film thickness d, nm: (5-1)–850, (5-2)–500, (5-3)–300, (5-4)–150 [49].

The Raman spectra was published earlier in [49].

[49] Lukyanova, L.N.; Bibik, A.Yu.; Aseev, V.A.; Usov, O.A.; Makarenko, I.V.; Petrov, V.N.; Nikonorov, N.V.; Kutasov, V.A. Surface morphology and Raman spectroscopy of thin layers of antimony and bismuth chalcogenides. Phys. Solid State. 2016, 58, 1440–1447. https://doi.org/10.1134/S1063783416070258

[50] Teweldebrhan, D.; Goyal, V.; Balandin, A.A. Exfoliation and characterization of bismuth telluride atomic quintuples and quasi-two-dimensional crystals. Nano Lett. 2010, 10, 1209-(18pp). https://doi.org/10.1021/nl903590b

[51] Plucinski, L.; Herdt, A.; Fahrendorf, S.; Bihlmayer, G.; Mussler, G.; Doring, S.; Kampmeier, J.; Matthes, F.; Burgler, D. E.; Grutzmacher, D.; Blugel, S.; Schneider, C.M. J. Appl. Phys. 2013, 113, 053706-(6pp). https://doi.org/10.1063/1.4789353

[52] Ovsyannikov, S.V.; Morozova, N.V.; Korobeinikov, I.V.; et al. Enhanced power factor and high-pressure effects in (Bi,Sb)2(Te,Se)3 thermoelectrics Appl. Phys. Lett. 2015, 106, 143901–5. https://doi.org/10.1063/1.4916947

[53] Korobeinikov, I.V.; Morozova, N.V.; Lukyanova, L.N.; Usov, O.A.; Ovsyannikov, S.V. On the Power Factor of Bismuth-Telluride-Based Alloys near Topological Phase Transitions at High Pressures. Semiconductors. 2019, 53, 732–736. https://doi.org/10.1134/S1063782619060083

Reviewer 2 Report

the comments are available in the attached files

Author Response

First of all, we would like to thank the Editor and the Reviewer for their constructive and insightful comments and suggestions to our manuscript. We modified the manuscript entitled “Thermal conductivity for p-(Bi, Sb)2Te3 films of topological insulators” as recommended. The modifications are summarized below.

Answers to Reviewer 2

Q1: The authors stated that “It is shown that in the p-Bi0.5Sb1.5Te3 films with low thermal conductivity the energy dependences of the relaxation time are enhanced, which is specific to the topological insulators”. While, this statement is not clear. The differences between topological insulators with topological semimetals/trivial insulators are not explained.

(Page 9-10)

The increase in |reff| observed in these films determines the enhancement of the relaxation time energy dependence (3) in the films of 3D-TIs [15,40,41]. The strong dependence of relaxation time on energy in TIs is related to the properties of the surface states of the Dirac fermions, which are protected by time-reversal symmetry from backscattering on non-magnetic defects. It leads to the formation of channels that are perfect for fast and ultralow dissipative electrical transport and improvement of thermoelectric properties [15]. The mechanism of Dirac fermions scattering and the dependence of τ(E) in TIs have been studied in [40,41] within the framework of non-equilibrium Boltzmann transport theory. Calculations of τ(E) were carried out for cases of long-range Coulomb scattering (LCS) and short-range delta-scattering (SDS), which in the limit gives dependences of τ(E) µ E and τ(E) µ E-1, respectively.

In the Dirac semimetal system such as graphene with the Fermi energy (EF) being located near the neutral Dirac point, the LCS dominates [40–42], and the relaxation time reaches the limit of τ(E) µ E. For 3D-TIs based on bismuth and antimony chalcogenides with low thermal conductivity, the limiting dependence of τ(E) µ E-1 corresponds to the SDS on neutral impurity or phonon scattering [40,41].

The reff values closest to the limiting τ(E) µ E-1 were obtained in an unannealed film of p-Bi0.5Sb1.5Te3 formed by discrete evaporation on a polyimide substrate (figure 6). Here, the reff varies from -0.925 to -0.77 depending on the reduced Fermi level η (figure 6, curve 1). According to scanning tunneling spectroscopy data [37], in the films of similar composition with energy gap of 240 meV, the Fermi energy relative to the conduction band edge is EF = 105 meV, and the Dirac point at ED =-130 meV is located close to the valence band edge [15,40,41]. In this case, the dependence of τ(E) becomes close to the limiting τ(E) µ E-1 [15,42]. The more noticeable difference of reff from -1 in the p–Bi0.5Sb1.5Te3 and p–Bi2Te3 films, as shown in figure 6, is assumed to be caused by the potential fluctuations arising from bulk carriers [15,42].

[15] Matsushita, S.Y.; Ichimura, K.; Huynh, K.K.; Tanigaki, K. Large thermopower in topological surface state of Sn-BSTS topological insulators: Thermoelectrics and energy-dependent relaxation times. Phys. Rev. Mater. 2021, 5, 014205–(8pp). https://doi.org/10.1103/PhysRevMaterials.5.014205

[40] Nomura, K.; MacDonald, A.H. Quantum Transport of Massless Dirac Fermions. Phys. Rev. Lett. 2007, 98, 076602-(4pp). https://doi.org/10.1103/PhysRevLett.98.076602

[41] Nomura, K.; Koshino, M.; Shinsei, R. Topological Delocalization of Two-Dimensional Massless Dirac Fermions. Phys. Rev. Lett. 2007, 99, 146806-(4pp).

https://doi.org/10.1103/PhysRevLett.99.146806

[42] Chiba, T.; Takahashi, S. Transport properties on an ionically disordered surface of topological insulators: Toward high-performance thermoelectrics. J. Appl. Phys. 2019, 126, 245704-(13pp). (2019). https://doi.org/10.1063/1.5131311

Q2: In the part 4 in page 7, the authors presented a formula to describe the electronic thermal conductivity, k # = L, r, η, σ T, and analyze the relation between the k# and the electrical conductivity σ . However, the effect of σ on k # are not provided both in experiments and theoretical explanations. Instead, the experimental data of Fermi function Fr+n (η), the effective scattering parameter reff and Lorentz number L versus η are presented, while the further discussions about significance of η are absent. I can’t understand the intention of this part.

The answer to question 2 is contained in the text:

κe = L(r, η) σT,

(page 9) The dependences Fr+n(η), reff(η) and α(η) (as shown in figures 5, 6, 7) illustrate the variation ranges of the reduced Fermi level η and the reff parameter corresponding to the experimental values of the thermopower coefficient α in the temperature range of 40-300 K.

(page 8) The parameter reff and the reduced Fermi level η were determined by the least squares method from the experimental values of the thermoelectric power coefficient α(r, η) for the studied films and the degeneracy parameter βd(r, η) in accordance with (2-5). The latter was calculated within the framework of the many-valley model of the energy spectrum from the ratios of isotropic factors in the expressions for electrical conductivity, Hall conductivity, and magnetoconductivity as discussed in [39]. The expressions for the thermoelectric power coefficient α(r, η) and the degeneracy parameter βd(r, η) in the isotropic relaxation time approximation (3).

(page 9) The dependences Fr+n(η), reff(η) and α(η) (as shown in figures 5, 6, 7) illustrate the variation ranges of the reduced Fermi level η and the reff parameter corresponding to the experimental values of the thermopower coefficient α in the temperature range of 40-300 K.

(page 10) The Lorentz number L(r, η) (1) employed in the calculation of κe was obtained using the data shown presented in figures 5–7.

(page 8) Figure 5. Fermi functions Fr+n(η) for the p-Bi0.5Sb1.5Te3 (1–4) and p-Bi2Te3 (5, 6) films, where n: 7–2.5, 8–1.5, 9– 0.5. Sample numbers in the figure 5 and subsequent figures correspond to table 1.

(page 9) Figure 6. The dependences of the effective scattering parameter of the charge carriers (reff) on the reduced Fermi level η in the p–Bi0.5Sb1.5Te3 (1–4) and p–Bi2Te3 (5, 6) films.

(page 10) Figure 7. The dependences of the thermoelectric power coefficient α (points 1–6 on curve 7) and the Lorentz number L (point 1–6 on curve 8) reduced Fermi level η in the p–Bi0.5Sb1.5Te3 (1–4) and p–Bi2Te3 (5, 6) films.

(page 16) The peculiarity of the dependence κe(T) in the temperature range of 80-110 K in the form of a hump is explained by the influence of potential fluctuations of bulk carriers (figure 11, inset). These fluctuations lead to a decrease in the effective scattering parameter |reff| with increasing of Fermi level due to the deviation of the energy dependence of relaxation time from the limiting τ(E) µE-1.

(page 9) The reff values closest to the limiting τ(E) µ E-1 were obtained in an unannealed film of p-Bi0.5Sb1.5Te3 formed by discrete evaporation on a polyimide substrate (figure 6). Here, the reff varies from -0.925 to -0.77 depending on the reduced Fermi level η (figure 6, curve 1). According to scanning tunneling spectroscopy data [37], in the films of similar composition with energy gap of 240 meV, the Fermi energy relative to the conduction band edge is EF = 105 meV, and the Dirac point at ED =-130 meV is located close to the valence band edge [15,40,41]. In this case, the dependence of τ(E) becomes close to the limiting τ(E) µ E-1 [15,42]. The more noticeable difference of reff from -1 in the p–Bi0.5Sb1.5Te3 and p–Bi2Te3 films, as shown in figure 6, is assumed to be caused by the potential fluctuations arising from bulk carriers [15,42]

(page 14, 15) The temperature dependences σ(T) (figure 11, inset) were analyzed using the r=1/σ, obtained as resistivity ratios between values of r measured at room (300 K) and low (20 K) temperatures in accordance with [29,55]. The observed decrease in the slopes σ(T) at low temperatures down to 80 K (figure 11, inset) turned out to be insufficient to change the σ(T) of the films from a metallic conductivity type to a semiconductor one, in contrast to [27], where the ratios of the r(10 K)/r(300 K) were considered.

Our studies have shown that the ratios of r(300 K)/r(20 K) (figure 11) as well as the total κ and the lattice κL thermal conductivities (figures 8 and 9) obtained in the p–Bi0.5Sb1.5Te3 are lower than in the p-Bi2Te3 regardless of the film formation technique used. A decrease in the ratios r(300 K)/r(20 K) in the p–Bi0.5Sb1.5Te3 films obtained by discrete evaporation is accompanied by a decrease in κ and κL (figures 8, 9, samples 1, 2, 4).

Q3: The Figure 11 provided the temperature-dependent electronic thermal conductivity ?#, which displayed a hump around 100 K without discussions and explanations.

The explanation about peculiarity of the dependence κe(T) in the range around 100 K is added to the text.

(Page 14-15)

The increase of |reff| with decreasing of Fermi level η (figure 6) is associated with the strengthening of electron scattering on intrinsic and impurity substitution defects at low temperatures [19,20]. Intensive scattering on defects with a temperature decrease from 80 to 40 K determines the reduction of electrical conductivity σ and corresponding decrease of electronic thermal conductivity κe. The "hump" on the κe(T) dependence observed in the temperature range of 80–110 K can be explained by potential fluctuations due to the influence of bulk carriers. These fluctuations result in deviation of the energy dependence of relaxation time from the limiting τ(E) µE-1 [15,40,41], and therefore to the decrease of the |reff| parameter with rise of η (figure 6).

[15] Matsushita, S.Y.; Ichimura, K.; Huynh, K.K.; Tanigaki, K. Large thermopower in topological surface state of Sn-BSTS topological insulators: Thermoelectrics and energy-dependent relaxation times. Phys. Rev. Mater. 2021, 5, 014205–(8pp). https://doi.org/10.1103/PhysRevMaterials.5.014205

[40] Nomura, K.; MacDonald, A.H. Quantum Transport of Massless Dirac Fermions. Phys. Rev. Lett. 2007, 98, 076602-(4pp). https://doi.org/10.1103/PhysRevLett.98.076602

[41] Nomura, K.; Koshino, M.; Shinsei, R. Topological Delocalization of Two-Dimensional Massless Dirac Fermions. Phys. Rev. Lett. 2007, 99, 146806-(4pp). https://doi.org/10.1103/PhysRevLett.99.146806

Q4: The authors stated that the “possibility of investigation of TSS in submicron thickness films” are confirmed by Figure 6. This conclusion lacks further explanations, leading reader’s confusion.

(Page 13)

      The dependence of thermal conductivity on the thickness of p-Bi0.5Sb1.5Te3 films (figure 10) associated with the influence of Dirac fermions surface states is consistent with micro-Raman spectroscopy studies [49]. The appearance of inactive longitudinal phonons A1u2 in the Raman spectra of p-Bi0.5Sb1.5Te3 films becomes noticeable at a film thickness of about 500 nm. The intensity of the A1u2 mode and the intensity ratios of the inactive longitudinal A1u2 and active transverse Eg2 optical phonon modes are found to increase with decreasing thickness [49]. Such a dependence of the intensities of longitudinal and transverse modes on the film thickness is determined by the distortion of the inversion symmetry of the crystal, and characterizes the effects associated with electronic surface states of Dirac fermions [50,51]. An inactive A1u2 mode of weak intensity was also observed in micro-Raman spectra of Bi2Te3 measured at pressures of about 3 GP [52], at which a topological surface phase transition was observed [53].

The properties of topological surface states of Dirac fermions in micron-thick Sn-(Bi,Sb)2(Te,S)3 films with tetradymite structure were investigated in thicker films [15, 54]. The estimations of the ratios of surface conductivity Gs to the total conductivity Gtot in the films showed that Gs/Gtot= 80% for film thicknesses of 3 µm at temperatures up to 210 K [15] and for 1 µm thick films at T up to 125 K [54], which confirms domination of surface transport properties.

[49] Lukyanova, L.N.; Bibik, A.Yu.; Aseev, V.A.; Usov, O.A.; Makarenko, I.V.; Petrov, V.N.; Nikonorov, N.V.; Kutasov, V.A. Surface morphology and Raman spectroscopy of thin layers of antimony and bismuth chalcogenides. Phys. Solid State. 2016, 58, 1440–1447.

https://doi.org/10.1134/S1063783416070258

[50] Teweldebrhan, D.; Goyal, V.; Balandin, A.A. Exfoliation and characterization of bismuth telluride atomic quintuples and quasi-two-dimensional crystals. Nano Lett. 2010, 10, 1209-(18pp). https://doi.org/10.1021/nl903590b

[51] Plucinski, L.; Herdt, A.; Fahrendorf, S.; Bihlmayer, G.; Mussler, G.; Doring, S.; Kampmeier, J.; Matthes, F.; Burgler, D. E.; Grutzmacher, D.; Blugel, S.; Schneider, C.M. J. Appl. Phys. 2013, 113, 053706-(6pp). https://doi.org/10.1063/1.4789353

[52] Ovsyannikov, S.V.; Morozova, N.V.; Korobeinikov, I.V.; et al. Enhanced power factor and high-pressure effects in (Bi,Sb)2(Te,Se)3 thermoelectrics Appl. Phys. Lett. 2015, 106, 143901–5. https://doi.org/10.1063/1.4916947

[53] Korobeinikov, I.V.; Morozova, N.V.; Lukyanova, L.N.; Usov, O.A.; Ovsyannikov, S.V. On the Power Factor of Bismuth-Telluride-Based Alloys near Topological Phase Transitions at High Pressures. Semiconductors. 2019, 53, 732–736. https://doi.org/10.1134/S1063782619060083

[54] Xu, Y.; Miotkowski, I.; Liu, C.; Tian, J.; Nam, H.; Alidoust, N.; Hu, J.; Shih, C.-K.; Hasan Z.; Chen, Y. P. Observation of topological surface state quantum Hall effect in an intrinsic three-dimensional topological insulator. Nat. Phys. 2014, 10, 956–963. https://doi.org/10.1038/nphys3140

In addition, the sentences in this manuscript are too excessively long and obscure to be read. The citation of figures with description in main text are mismatched, such as Fig. 10 and Fig.11 in page 11. Several parameters directly take the values without provenance or origin, such as the parameter n in page 7.

English has been checked and improved throughout the manuscript. Some of long and complex sentences were improved for better reading.

Page 2, paragraph 2, 4. Page 3, paragraph 2. Page 8, paragraph 2. Page 12, paragraph 2, 3.

Page 13, paragraph 3. Page 16, paragraph 2, 4 (conclusion)

Figure 10 was replaced by figure 11.

(Page 8)

The parameter n in the expression Fr + n(η) determines the variation range of reff.

Reviewer 3 Report

In this manuscript, the authors discussed the investigation of the thermal conductivities of films of topological insulators p–Bi0.5Sb1.5Te3 and p–Bi2Te3, focusing on the temperature dependence of total, crystal lattice, and electronic thermal conductivities. The study showed that there is a correlation between the thermal conductivity and the morphology of the interlayer surface in the films studied. It's a complete work given the adequate results, and I am favorable to the publication of this manuscript.

Author Response

Answer to Reviewer 3

Dear Reviewer,

We are very pleased that the results of the study satisfied you and are grateful for the recommendation to publish the manuscript in the Magnetochemistry journal.

Reviewer 4 Report

Ref: Magnetochemistry 2022, 8, x. https://doi.org/10.3390/xxxxx

Journal: Magnetochemistry

Authors: Lidia N. Lukyanova, Yuri A. Boikov, Oleg A. Usov, Viacheslav A. Danilov and Igor V. Makarenko, Vasilii N. Petrov

Dear authors I have few comments which may help to improve the quality of the manuscript regarding readability and understanding for the audience.

1.      Kindly elaborate on the experimental methodology in more detail, clearly specifying the controlled experimental conditions etc. For instance provide further detail on the methods of discrete and thermal evaporation and the thermoelectric properties of the formed films were measured using the Physical Property Measurement System (PPMS) Thermal Transport Option experimental setup.

2.     The authors write, “To describe the structure of Bi2Te3 and its solid solutions, a primitive rhombohedral or hexagonal unit cell is used. The hexagonal unit cell of the Bi2Te3 has a and c parameters of 4.385 Å and 30.487 Å , respectively”.

How did the authors probe the structure for the parameters? Please elaborate.

3.     As the main target of the research is to enhance the figure of merit Z which can be achieved by reducing the thermal conductivity and increasing the power factor/electrical conductivity, the suitable title of the manuscript may be like "On the Enhancement of Figure of Merit of.....".

4.     As the Z is a complex function of thermal and electrical conductivity, and Seebeck coefficient S, could the authors obtain results for S or power factor by showing its relation on Z for further evidence to support their conclusion. 

5.     How the authors could interpret the measured thermoelectric properties of the materials to the general Boltzmann theory? See for example reference International Journal of Thermophysics (2019) 40:104; https://doi.org/10.1007/s10765-019-2572-7.

6.       There is a considerable overlapping of text with the published articles, so please reduce the similarity index to further enhance the originality of the manuscript.

Best regards

Author Response

First of all, we would like to thank the Editor and the Reviewer for their constructive and insightful comments and suggestions to our manuscript. We modified the manuscript entitled “Thermal conductivity for p-(Bi, Sb)2Te3 films of topological insulators” as recommended. The modifications are summarized below.

Answers to Reviewer 4

  1. Kindly elaborate on the experimental methodology in more detail, clearly specifying the controlled experimental conditions etc. For instance provide further detail on the methods of discrete and thermal evaporation and the thermoelectric properties of the formed films were measured using the Physical Property Measurement System (PPMS) Thermal Transport Option experimental setup.

page 3

            The polycrystalline films of solid solutions p–(Bi,Sb)2Te3 and p–Bi2Te3 were obtained by the methods of discrete and thermal evaporation in an isothermal chamber which provides a vacuum of 1 10-6 torr and homogeneous temperature distribution along substrate plane. Fresh muscovite mica cleavage planes with a thickness of 3-10 µm and polyimide films with a thickness of 6-20 µm were used as substrates. To obtain the films by the thermal evaporation method, the starting material was loaded into a quartz crucible heated by molybdenum coil. The films deposition rate was 10-15 Å/s. During the films formation by discrete evaporation, the initial material in the form of powder with a grain size of about 10 µm was passed by small portions into heated quartz crucible, where it instantly evaporated.

page 7

            The thermoelectric properties of the formed films were measured using the Physical Property Measurement System (PPMS) [40] that controlled by Quantum Design Thermal Transport Option (TTO) software. Thermal conductivity of a material was measured by TTO system as the temperature drop along the sample as a known quantity of heat that passes through the sample. The Seebeck coefficient was obtained by TTO as an electrical voltage drop caused by temperature gradient across the studied sample. The electrical resistivity measurements were carried out by the PPMS system using the standard four-probe method. The PPMS allows for simultaneous measurements of thermoelectric properties over a wide temperature range from 2 to 390 K with a temperature change rate of ±0.5 K/min. The typical measurement accuracy of the Seebeck coefficient and thermal conductivity is ±5%, and typical precision of electrical resistivity is 0.01% for 1 Ω and 200 µA.

  1. User’s Manual. Quantum Design. Physical Property Measurement System Thermal Transport Option. 2002, Part Number 1684-100B, 1–77.

  1. The authors write, “To describe the structure of Bi2Te3 and its solid solutions, a primitive rhombohedral or hexagonal unit cell is used. The hexagonal unit cell of the Bi2Te3 has a and c parameters of 4.385 Å and 30.487 Å , respectively”.

How did the authors probe the structure for the parameters? Please elaborate.

page 4

According to X-ray powder difraction data the hexagonal unit cell of the Bi2Te3 has a and c parameters of 4.3805 Å and 30.487 Å, respectively [31].

  1. Francombe, M.H. Structure-cell data and expansion coefficients of bismuth telluride. 1958. Brit. J. Appl. Phys. 9, 415–417. https://doi.org/10.1088/0508-3443/9/10/307

3. As the main target of the research is to enhance the figure of merit Z which can be achieved by reducing the thermal conductivity and increasing the power factor/electrical conductivity, the suitable title of the manuscript may be like "On the Enhancement of Figure of Merit of.....".

4. As the Z is a complex function of thermal and electrical conductivity, and Seebeck coefficient S, could the authors obtain results for S or power factor by showing its relation on Z for further evidence to support their conclusion. 

            Our article is devoted to the study of the thermal conductivity of films obtained under different technological parameters. Data on the thermoelectric efficiency in similar films will be used for another studies.

Figure 1. Temperature dependence of the Seebeck coefficient S in p–Bi0.5Sb1.5Te3 films.

Figure 2. Temperature dependence of electrical conductivity σ in p–Bi0.5Sb1.5Te3 films.

Figure 3. Temperature dependence of the power factor S2σ in p–Bi0.5Sb1.5Te3 films.

Figure 4. Temperature dependence of the figure of merit Z in p–Bi0.5Sb1.5Te3 films.

Temperature dependences of the Seebeck coefficient, electrical conductivity, power factor, and thermoelectric efficiency in p–Bi0.5Sb1.5Te3 films are shown in figures 1-4.

The technological parameters description of the p–Bi0.5Sb1.5Te3 films:

1 - discrete evaporation, polyimide, unannealed.

2 - discrete evaporation, polyimide, annealed.

3 - discrete evaporation, muscovite, annealed.

4 - thermal evaporation, polyimide, unannealed.

         The highest figure of merit Z in the temperature range of 200-300 K was achieved in annealed films 2 and 3, formed by discrete evaporation on a polyimide (2) and mica (3) substrates. Similar Z values in these p–Bi0.5Sb1.5Te3 films at T=200-300 K are determined by the increase in the power factor in film 3 compared to film 2, while the thermal conductivity in the film 3 is higher than in film 2. In film 1, which has the lowest thermal conductivity, the power factor is also low due to reduced electrical conductivity.

  1. How the authors could interpret the measured thermoelectric properties of the materials to the general Boltzmann theory? See for example reference International Journal of Thermophysics (2019) 40:104; https://doi.org/10.1007/s10765-019-2572-7.

         In the article (https://doi.org/10.1007/S10765-019-2572-7) the density functional theory (DFT) formalism were used to calculate the structural, electronic, magnetic, and thermoelectric properties of thorium monopnictides ThPn (Pn = N, P, As) under normal environmental conditions. The calculations based on the Boltzmann transport theory in the constant relaxation time approximation of the thermoelectric properties of the ThPn compounds in the temperature range of 300-500 K exhibit large values of electrical conductivity, Seebeck coefficient, and thermoelectric power factor. However, due to the high thermal conductivity, the thermoelectric efficiency is low. The most of thorium monopnictides are compounds with a cubic crystal structure with a zero band gap.

         Thermoelectrics based on bismuth and antimony chalcogenides (p–Bi0.5Sb1.5Te3) considered in our manuscript are the van der Waals layered crystals with a tetradymite structure [4] with layered hexagonal unit cell. Such structures are known to easily cleaves along the cleavage planes (0001). These thermoelectrics are 3D topological insulators. The topological surface states (TSS) arise due to inversion of the energy gap edges caused by strong spin-orbit interaction [3–5]. The band gap of p–Bi0.5Sb1.5Te3 is about Eg = 0.25 meV. One of the distinctive feature of these materials is a strong energy dependence of the relaxation time [15, 42,43].

            So, bismuth and antimony chalcogenides are significantly different from thorium monopnictides ThPn (Pn = N, P, As). Therefore, we did not find additional information in the mentioned example (https://doi.org/10.1007/S10765-019-2572-7), despite the DFT calculations and analysis of the thermoelectric properties of ThPn according to the Boltzmann theory.

            Our calculations of the Lorenz number, carried out taking into account the effective scattering parameter that determines the energy dependence of the relaxation time of charge carriers, are in good agreement with the articles [42,43]. See pages 9, 10 of the manuscript:

            The strong dependence of relaxation time on energy in TIs is related to the properties of the surface states of the Dirac fermions, which are protected by time-reversal symmetry from backscattering on non-magnetic defects. It leads to the formation of channels that are perfect for fast and ultralow dissipative electrical transport and improvement of thermoelectric properties [15]. The mechanism of Dirac fermions scattering and the dependence of τ(E) in TIs have been studied in [42,43] within the framework of non-equilibrium Boltzmann transport theory. Calculations of τ(E) were carried out for cases of long-range Coulomb scattering (LCS) and short-range delta-scattering (SDS), which in the limit gives dependences of τ(E) µ E and τ(E) µ E-1, respectively.

            In the Dirac semimetal system such as graphene with the Fermi energy (EF) being located near the neutral Dirac point, the LCS dominates [42–44], and the relaxation time reaches the limit of τ(E) µ E. For 3D-TIs based on bismuth and antimony chalcogenides with low thermal conductivity, the limiting dependence of τ(E) µ E-1 corresponds to the SDS on neutral impurity or phonon scattering [42,43].

            The reff values closest to the limiting τ(E) µ E-1 were obtained in an unannealed film of p-Bi0.5Sb1.5Te3 formed by discrete evaporation on a polyimide substrate (figure 6). Here, the reff varies from -0.925 to -0.77 depending on the reduced Fermi level η (figure 6, curve 1). According to scanning tunneling spectroscopy data [38], in the films of similar composition with energy gap of 240 meV, the Fermi energy relative to the conduction band edge is EF = 105 meV, and the Dirac point at ED =-130 meV is located close to the valence band edge [15,42,43]. In this case, the dependence of τ(E) becomes close to the limiting τ(E) µ E-1 [15,44]. The more noticeable difference of reff from -1 in the p–Bi0.5Sb1.5Te3 and p–Bi2Te3 films, as shown in figure 6, is assumed to be caused by the potential fluctuations arising from bulk carriers [15,44].

References

3. Gilbert, M.J. Topological electronics. Comm. Phys. 2021, 4, 70–(12pp). https://doi.org/10.1038/s42005-021-00569-5

4. Heremans, J.; Cava, R.; Samarth, N. Tetradymites as thermoelectrics and topological insulators. Nature Rev. Mater. 2017, 2 17049–(21pp). https://doi.org/10.1038/natrevmats.2017.49

5. Ngabonziza, P. Quantum transport and potential of topological states for thermoelectricity in Bi2Te3 thin films. Nanotechnology. 2022, 33, 192001–(22pp). https://doi.org/10.1088/1361-6528/ac4f17

42. Nomura, K.; MacDonald, A.H. Quantum Transport of Massless Dirac Fermions. Phys. Rev. Lett. 2007, 98, 076602–(4pp). https://doi.org/10.1103/PhysRevLett.98.076602

43. Nomura, K.; Koshino, M.; Shinsei, R. Topological Delocalization of Two-Dimensional Massless Dirac Fermions. Phys. Rev. Lett.2007, 99, 146806-(4pp). https://doi.org/10.1103/PhysRevLett.99.146806

44. Chiba, T.; Takahashi, S. Transport properties on an ionically disordered surface of topological insulators: Toward high-performance thermoelectrics. J. Appl. Phys. 2019, 126, 245704-(13pp). https://doi.org/10.1063/1.5131311

38. Lukyanova, L.N.; Makarenko, I.V.; Usov, O.A.; Dementev, P.A. Scanning tunneling spectroscopy of the surface states of Dirac fermions in thermoelectrics based on bismuth telluride. Semicond. Sci. Technol. 2018, 33, 055001–(9pp). https://doi.org/10.1088/1361-6641/aab538

15. Matsushita, S.Y.; Ichimura, K.; Huynh, K.K.; Tanigaki, K. Large thermopower in topological surface state of Sn-BSTS topological insulators: Thermoelectrics and energy-dependent relaxation times. Phys. Rev. Mater. 2021, 5, 014205–(8pp). https://doi.org/10.1103/PhysRevMaterials.5.014205

There is a considerable overlapping of text with the published articles, so please reduce the similarity index to further enhance the originality of the manuscript.

            We have checked the similarity index of our manuscript using plagiarism checking system and found quite low overlapping of text with the published articles excluding the preprint of our manuscript posted to the "arXiv" (https://arxiv.org/abs/2207.04477) open-access free distribution service. We also found some overlapping with a number of articles related mainly to appropriate terminology of topological physical phenomena.

            So, we have made some corrections to the text of our manuscript in order to enhance the originality of the text. The corrected sentences are highlighted by color in the manuscript on pages 1, 2, 4, 6, 7, 16.

Round 2

Reviewer 1 Report

  The Authors have adequately addressed all of my comments.  I have no additional or further comments, and can recommend acceptance of the manuscript in its current form.

Author Response

Answer to Reviewer 1

Dear Reviewer,

We are very pleased that our answers have fully satisfied you and are grateful for the constructive and insightful comments and suggestions that have helped to improve our manuscript.

Reviewer 2 Report

The reply does not respond to the questions raised from the referee  in a proper way. The details not explained clearly are directly related the understanding of the work and the physics underneath, thus, the paper is not recommended for publication.

Author Response

Answer to Reviewer 2

Dear Reviewer,

We are sorry that our answers did not satisfy you, but we appreciate the constructive and insightful comments and suggestions that have helped to improve our manuscript.

Reviewer 4 Report

Dear authors thank you for your response. Although manuscript has been improved, yet I find considerable overlapping with earlier work with about 7% from only a single source which is serious issue for academic writing. This issue must be addressed by furnishing plagiarism by excluding the references. 

Author Response

Thank you for your comments and suggestions. All modifications in the manuscript are highlighted in blue.

Answer to Reviewer 4

         We have received the originality report of our manuscript from section managing Editor. We have compared both texts and didn't find any coincidence or overlapping between our manuscript and presented article Szczech, J.R.; Higgins, J.M.; Jin, S. Enhancement of the thermoelectric properties in nanoscale and nanostructured materials. J. Mater. Chem. 2011, 21, 4037–4055. https://doi.org/10.1039/c0jm02755c.

         This article is devoted to the study of various nanoscale and nanostructured thermoelectric materials relative to their corresponding bulk materials which were shown to be promising for ZT enhancement. This article does not discuss the topological properties, the energy dependence of the relaxation time of charge carriers and the methods of obtaining films by discrete and thermal evaporation. The overlapping of 8% (labeled by "1" and highlighted in red in the text of the originality report) is related to our first-version of manuscript posted to the "arXiv" (https://arxiv.org/abs/2207.04477) open-access free distribution service as we already wrote in the previous answers.

         The link 1 in the originality report follows to the page of https://www.researchgate.net/publication/255749411_Enhancement_of_the_thermoelectric_properties_in_nanoscale_and_nanostructured_materials concerning the article Szczech, J.R.; et. al. J. Mater. Chem. 2011, which contains only author information, affiliation and abstract without full article text. Lower on this page of https://www.researchgate.net link one can find the references to preprint of our manuscript posted to the "arXiv" (https://arxiv.org/abs/2207.04477). This reference to the preprint contains a few paragraphs of our manuscript which affect the results of automatic plagiarism checking, which in turn results in noticeable distortion and produces overestimation of text similarity index.

         In the considered manuscript as well as in other articles we have used the common terms, phrases and terminology to describe considered phenomena and topological properties provided, if necessary, with appropriate references.

         We have corrected some sentences in the manuscript to enhance the text originality. The corrected sentences are highlighted by color in the text on pages 1, 2, 7, 9.

         We hope the revised manuscript will meet the high standards and the requirements of academic publishing in Magnetochemistry journal.
